# Continued preference for suboptimal habitat reduces bat survival with white-nose syndrome

Skylar R. Hopkins [1,2 ✉], Joseph R. Hoyt [1], J. Paul White[3], Heather M. Kaarakka[3], Jennifer A. Redell[3], John E. DePue[4], William H. Scullon[5], A. Marm Kilpatrick[6] & Kate E. Langwig [1]

Habitat alteration can influence suitability, creating ecological traps where habitat preference and fitness are mismatched. Despite their importance, ecological traps are notoriously difficult to identify and their impact on host–pathogen dynamics remains largely unexplored. Here we assess individual bat survival and habitat preferences in the midwestern United States before, during, and after the invasion of the fungal pathogen that causes white-nose syndrome. Despite strong selection pressures, most hosts continued to select habitats where disease severity was highest and survival was lowest, causing continued population declines. However, some individuals used refugia where survival was higher. Over time, a higher proportion of the total population used refugia than before pathogen arrival. Our results demonstrate that host preferences for habitats with high disease-induced mortality can create ecological traps that threaten populations, even in the presence of accessible refugia.

[1] Department of Biological Sciences, Virginia Tech, Blacksburg, VA 24060, USA. [2] Department of Applied Ecology, North Carolina State University, Raleigh, NC 27695, USA. [3] Wisconsin Department of Natural Resources, Bureau of Natural Heritage Conservation, Madison, WI 53703, USA. [4] Michigan Department of Natural Resources, Baraga, MI 49870, USA. [5] Michigan Department of Natural Resources, Norway, MI 49908, USA. [6] Department of Ecology and Evolutionary Biology, University of California, Santa Cruz, CA 95060, USA. ✉email: skylar.hopkins@vt.edu

**M**etapopulations are frequently distributed across a patchwork of sites, where source and sink subpopulations occupy patches with different habitat quality[1,2]. If habitat quality is affected by habitat alteration, such as the spread of an invasive species, native population dynamics can shift[3]. At one extreme, a population may be extirpated if an invasive species reduces individual survival or reproduction in all available habitats. In contrast, native species may avoid or at least delay extinction if habitat refugia exist. In these refugia, individual survival or reproduction remain high because invasive species cannot colonize or species interactions are mediated by other biotic or abiotic interactions[4,5]. However, even when sufficient refugia exist, the population may still decline or go extinct if behavioral cues for preferred habitat become decoupled from actual suitability[6], such that individuals show greater or equal preference for habitats with low survival or reproduction, known as ecological traps (Fig. 1)[7]. Ecological traps are notoriously difficult to identify because they require individual-based data[7], which are challenging to collect for free-ranging species[8]. Identifying traps is critical because they require different management strategies to maximize long-term metapopulation persistence during and after invasions than strategies used for unpreferred sink habitats or refugia[3,9].

Invasive pathogens have been particularly difficult to control, as evidenced by the rapid and unchecked distribution shifts, declines, and extinctions that they have caused for native wildlife (e.g., chytridiomycosis in amphibians, avian malaria in Hawaiian honeycreepers, white-nose syndrome in bats, withering syndrome in abalone, etc.)[10–12]. Potential host species frequently have life history traits that are influenced by temperature and vary along thermal gradients. For many emerging infectious disease systems, including those listed above, interactions between the invasive pathogen and native hosts are also temperature-dependent, where temperature can affect pathogen transmission[13,14], host resistance or tolerance to infection (e.g., "behavioral fever")[15,16], or pathogen growth rates[17–19]. Given temperature-dependent interactions between hosts and pathogens, host survival or reproduction across temperature gradients may shift following pathogen invasion. Changes in habitat suitability caused by pathogen invasion may or may not cause hosts' thermal preferences to change via learning or selection (Fig. 1). However, host temperature preferences and habitat suitability are rarely known before pathogen invasion or quantified after invasion, making it difficult to identify and manage thermal traps, sinks, and refugia for host species. Failure to successfully distinguish these post-invasion habitat patches could create management traps, where ecological traps are mistakenly prioritized for conservation efforts[20,21].

Several bat species in North America have recently experienced severe population declines[22–25] due to a fungal pathogen introduced from Eurasia, *Pseudogymnoascus destructans*, which causes white-nose syndrome[26–29]. The fungus infects bats and grows into their epidermal tissue during hibernation, when bats cool their bodies down to near ambient hibernaculum temperatures. Ambient temperatures vary within and among hibernacula, and bats in eastern North America roost between 7 °C and 11 °C on average, depending on the bat species[30,31]. Individual bats can select microclimates that optimize temperature-dependent energy use with other physiological constraints during hibernation[32–35], and over the range of temperatures used by bats, the fungus has nonlinear, temperature-dependent growth in vitro[17]. In particular, the fungus grows optimally near 12–16 °C on growth media, with a thermal range between 0 °C and ~21 °C[17]. Fungal loads on bat wings are strongly correlated with damage to wing tissue[36] and population impacts[25,37]. Therefore, we hypothesized that the invasion of *P. destructans* would alter thermal habitat suitability for bats, where warmer roosts would have higher fungal growth and bat mortality, and bats might alter their microclimate preferences across hibernacula from pre-invasion to post-invasion in response to this selection pressure.

To test this hypothesis, we conduct a longitudinal and cross-sectional study before, during, and after fungal invasion in 22 sites in the Midwestern U.S. to quantify how changes in fungal loads and recapture rates of little brown bats (*Myotis lucifugus*) during hibernation are mediated by roosting temperatures. We then assess whether temperature-mediated disease impacts cause little brown bat roosting temperatures to change from pre-fungal to post-fungal invasion and whether bats continue to prefer thermal trap microclimates where disease impacts are highest. We find that over winter increases in fungal loads are highest and recapture rates are lowest for bats roosting at warm temperatures during early hibernation. At the hibernaculum level, over winter

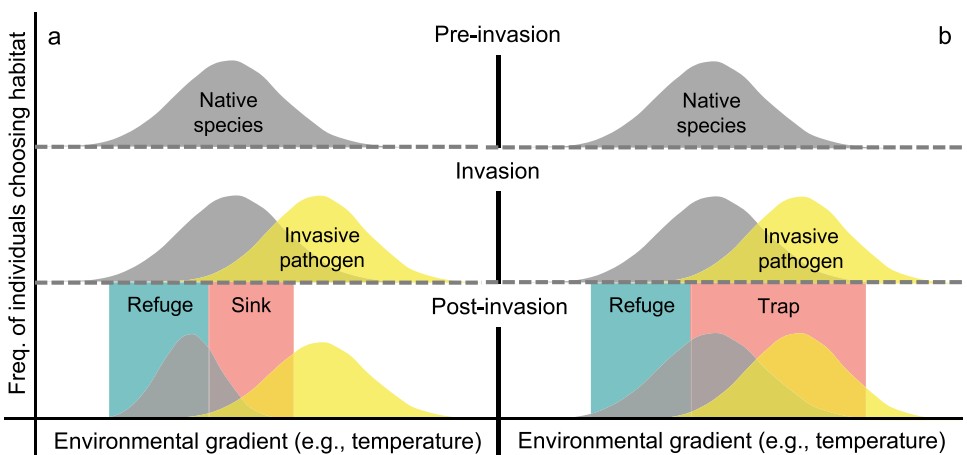

**Fig. 1 Habitat refugia, sinks, and traps for native species.** Before invasion, individuals are distributed across environmental gradients such that habitat preferences generally match habitat suitability. Invasive species can cause some or all habitats along an environmental gradient to become unsuitable for native species' survival or reproduction, and (**a**) individual preferences may change with suitability due to learning or selection. However, (**b**) native species may also continue to prefer unsuitable habitats, and these habitats will become ecological traps after invasion. Whether preferences change or not, native species' post-invasion distributions might be shifted towards habitat refugia where invasive species cannot colonize or negative species interactions are sufficiently mediated by environmental conditions.

| Sample units | Sample Sizes | | | Excluded Units | Analysis |
|---|---|---|---|---|---|
| | Pre-Invasion | Invasion | Post-Invasion | | |
| Bats counted in 12 sites surveyed during all three invasion periods | n=6138 | n=2034 | n=301 | | Distribution shift analysis (N=8473) |
| Sampled bats from 12 sites + Sampled bats from 10 sites | n=455 / n=118 | n=439 / n=82 | n=176 / n=2 | Only 25 bats per site were sampled per survey to limit potential disturbance to sensitive vertebrate species. | |
| Infected bats | n=11 | n=459 | n=231 | Uninfected bats were not used in the recapture analysis in order to control for timing of infection as much as possible. | |
| Banded, infected bats | n=3 | n=98 | n=158 | | Recapture analysis (N=259) |
| Recaptured bats | n=1 | n=39 | n=83 | Bats that were not recaptured were not used in the fungal load change analysis because we did not have a second time point for comparison. | Fungal load change analysis (N=123) |

**Fig. 2 Survey design and sample sizes for analyses.** We surveyed 12 hibernacula before invasion, during invasion (1–2 years since detection), and after invasion (>2 years since detection), and we surveyed 10 other hibernacula during at least one of the invasion periods. Bats counted in the 12 long-term sites were included in our analysis of how bat temperature distributions in November shifted from pre-invasion to post-invasion. After counting bats, we sampled up to 25 bats per site per survey, stratified by section, to quantify individual fungal loads and early hibernation roosting temperatures. If these sampled bats could be reached to safely remove them briefly from their roosts, we also banded them. All banded bats that were infected during early hibernation were included in our recapture analysis, and all recaptured bats that were infected during early hibernation were included in our fungal load change analysis. Note that only a few bats were infected and banded during Year 0, the year that the fungus was first detected at a given site. We included Year 0 as pre-invasion because bats do not suffer severe disease impacts or mortality in Year 0 (Hoyt et al.)[38]. All sample sizes represent the number of bats sampled during early hibernation.

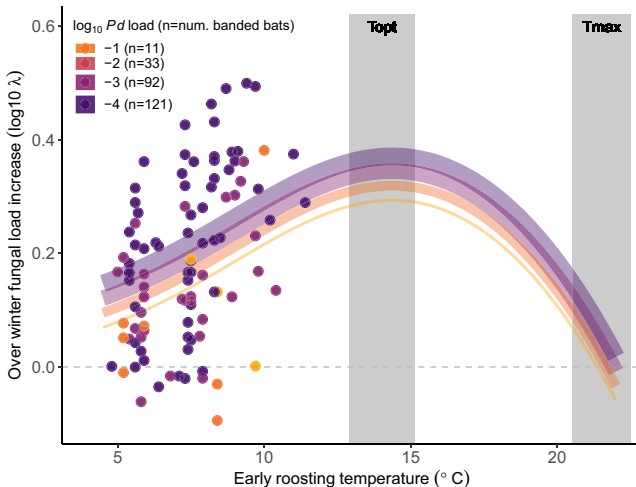

**Fig. 3 Temperature and changes in fungal loads on bats.** Observed fungal load changes on individual infected bats that were recaptured in late hibernation ($N = 123$), where the curves depict the fitted Logan-10 temperature-dependent growth curve with fungal-load-dependent intercepts. The shaded regions indicate the 95% credible intervals for the $T_{opt}$ and $T_{max}$ parameter estimates from the prior laboratory experiment.

population growth rates decline with increasing average roosting temperature, whereas over summer population growth rates tend to increase with temperature, suggesting that bats preferentially immigrate to the warmer sites with the highest mortality. Furthermore, across 12 hibernacula surveyed before, during, and after invasion, half of the metapopulation still roosts in relatively warm microclimates post-invasion, even though cooler refuges are available.

## Results

**Over winter changes in fungal loads on bats**. On individual, hibernating little brown bats that were banded and infected during early hibernation and recaptured in late hibernation ($N = 123$, Fig. 2), over winter changes in fungal loads increased with early hibernation roosting temperatures (Spearman's rho = 0.42, $p$ value < 0.0001) and decreased with early fungal loads (Spearman's rho = 0.40, $p$ value < 0.0001). A Logan-10 nonlinear growth model with a flexible, fungal load-dependent intercept also supported larger changes in fungal loads with increasing roosting temperature and explained 29% of the variation in observed changes in fungal loads (Fig. 3).

**Bat recapture rates**. Correspondingly, recapture probabilities for infected bats that were sampled in early hibernation ($N = 259$) decreased sharply with early hibernation roosting temperatures when bats had relatively low to moderate fungal loads during early hibernation (Logistic GLMM intercept ± SE = $-4.57 \pm 3.53$; interaction term = $-0.28 \pm 0.13$, $p = 0.03$; site-level random variance = 1.02) (Fig. 4). In contrast, recapture probabilities were consistently low for bats with relatively high fungal loads across all early hibernation temperatures (main temperature effect = $0.34 \pm 0.42$, $p = 0.42$; main fungal load effect = $-2.80 \pm 1.08$, $p = 0.0099$) (Fig. 4). This model accurately predicted 73.2% of bat recaptures (5-fold cross-validation; 95% bootstrapped confidence interval for AUC = 71.4–74.4%) (Supplementary Fig. 6), which was substantially better than the intercept-only model with random site effects (46%; the 95% bootstrapped confidence interval for AUC overlapped 50%, as expected: 40.3–52.6%).

**Bat distribution shifts**. Across 12 hibernacula that were surveyed before ($N = 455$ sampled and 6138 counted bats), during ($N = 439$ sampled and 2034 counted bats), and after ($N = 176$ sampled

and 301 counted bats) fungal invasion, a large proportion (52%) of the few bats persisting in the region during post-invasion years still used relatively warm roosts (i.e., >8 °C). Warmer hibernacula also tended to have higher immigration rates over the summer (between March and November surveys), as indicated by a marginal interaction between season (over summer versus over winter) and early hibernation temperatures (Gaussian LM interaction term = 0.14 ± 0.08, $p = 0.08$; Supplementary Discussion, Supplementary Fig. 8). However, the average observed early hibernation roosting temperature used by bats was ~1 °C lower during post-invasion than during pre-invasion (pre-invasion mean ± SE = 8.95 ± 0.02; pre-invasion to post-invasion change =

−0.85 ± 0.10; $p < 0.001$) (Fig. 5B, Fig. 6). This change was not attributable to interannual variation in hibernacula temperatures, because there was no trend in average hibernacula temperatures from pre-invasion to post-invasion as measured by temperature loggers in fixed locations (Supplementary Fig. 7). Instead, the average change in roosting temperatures occurred because over winter population losses (mortality and potential emigration) were highest in warm sites during post-invasion years (Fig. 4, Supplementary Discussion, Supplementary Fig. 8). Thousands of bats died in the warmest sites with the largest pre-invasion colonies, which caused a proportional shift toward cooler roosting distributions over time (Fig. 6), even though bats still preferentially used warm roosts.

The average decline in early hibernation roosting temperatures could explain why bat recapture probabilities increased over the invasion period (logistic GLMM; intercept ± SE = −1.73 ± 0.62; slope = 0.50 ± 0.18, $p = 0.005$; site random effect variance = 1.49) (Fig. 5a). In particular, after accounting for early hibernation roosting temperatures and fungal loads in a partial regression analysis, there was no relationship between recapture probabilities and invasion year (partial regression intercept = 0.0015 ± 0.06; slope = 0.03 ± 0.10, $p = 0.77$) (Fig. 5c).

## Discussion

As *P. destructans* spread across the Midwestern U. S., causing major population declines[38], the probability that persisting little brown bats survived their infections increased each winter after the initial epidemic (Fig. 5a). Bats may be evolving host-specific mechanisms to persist with *P. destructans*[39–42], but here we show that increased bat survival could be explained by environmentally-mediated disease outcomes. Fungal growth rates were highest and recapture rates and over winter population growth rates were lowest for bats that roosted at relatively warm temperatures during early hibernation. The pathogen was present in colder roosts, but changes in fungal loads were smaller and bat recapture rates were higher in these thermal refugia. Overall, differential population declines between relatively warm hibernacula and thermal refugia caused average roosting temperatures across our sites to decline by

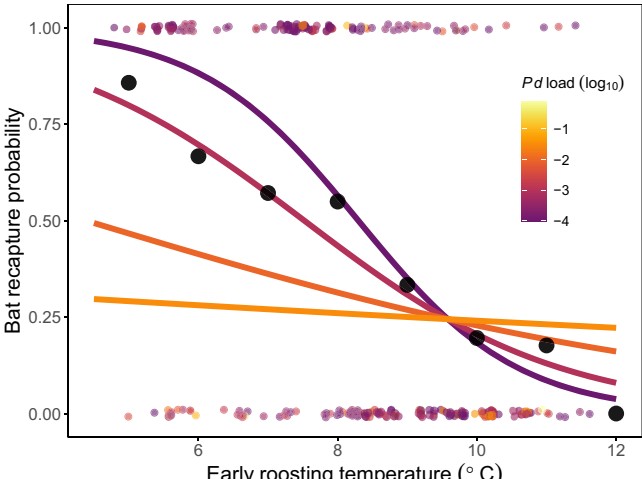

**Fig. 4 Temperature and bat recapture probabilities.** The probability that a banded little brown bat that was infected during early hibernation ($N = 259$) was recaptured in late hibernation. Small points show the outcomes for individual bats (1 = recaptured, 0 = not recaptured) and black points show the proportion of recaptured bats for each temperature, binned by 1 °C intervals.

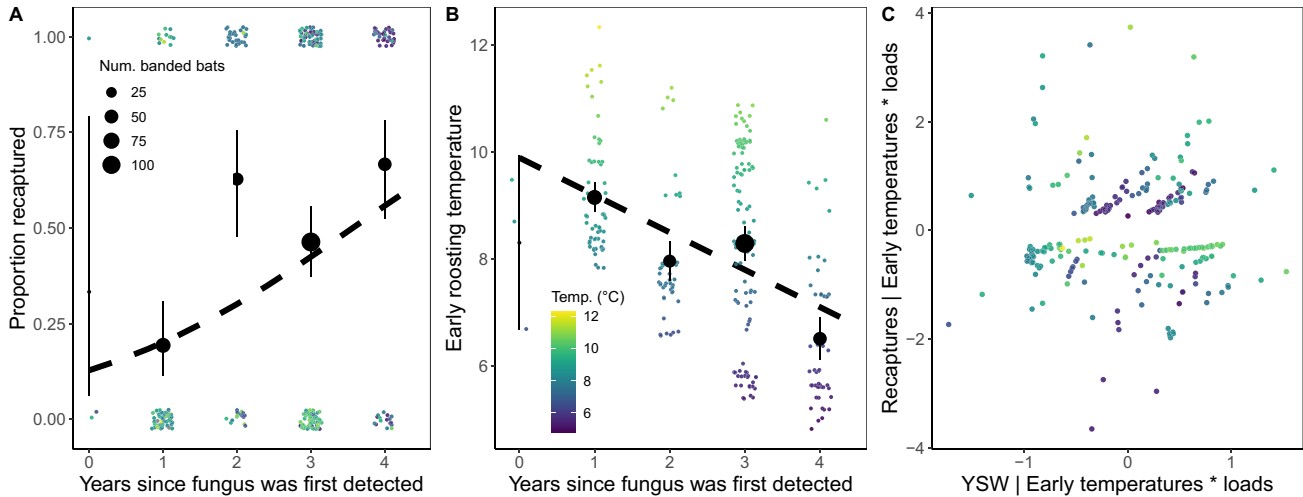

**Fig. 5 Partial regression of recapture outcomes and invasion year. a** Years since the detection of *P. destructans* at a site and proportion of infected, banded little brown bats ($n = 259$) that were recaptured (black points +/−95% Wilson binomial confidence intervals). The small points show the outcomes for individual bats (1 = recaptured, 0 = not recaptured) and the dashed line shows the best fitting binomial GLM without accounting for changes in roosting temperature over time. **b** Years since the detection of *P. destructans* at a site and average early hibernation bat roosting temperatures ($n = 259$ bats, black points +/−95% Gaussian confidence intervals), which significantly negatively decreased across invasion years (also see Fig. 5). **c** Partial regression analysis of years since detection of the fungus (YSW) and bat recapture probabilities after accounting for early hibernation temperatures and fungal loads. After accounting for the change in bat roosting temperatures throughout invasion (**b**), there was no additional significant effect of fungal invasion year on recapture probabilities (**c**).

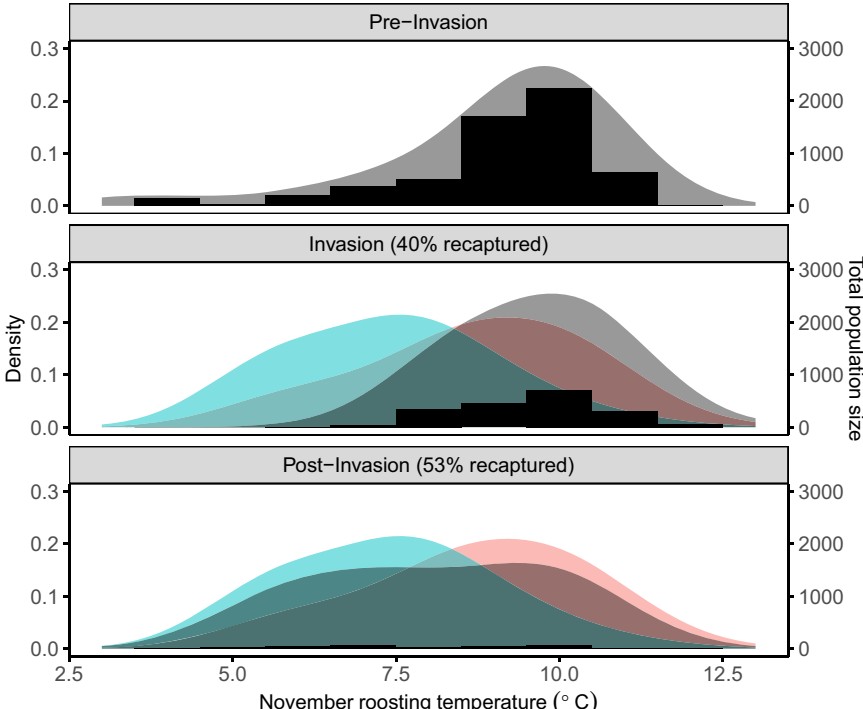

**Fig. 6 Changes in bat roosting temperatures during pathogen invasion.** The gray density plots (left vertical axis) show the sampled and estimated early hibernation temperature distributions of all counted little brown bats. The blue and red density plots (left vertical axis) show the early hibernation temperature distributions of banded bats that were (blue) and were not (red) recaptured during late hibernation. The gray histogram repeats the gray density plot on the count scale (right vertical axis), to illustrate the large population decline from pre-invasion to post-invasion, especially in warmer microsites. The parentheticals show the increase in banded bat recapture rates across invasion periods (also see Fig. 4a). Overall, the average little brown bat roosting temperature in early hibernation declined ~1 °C from pre-fungal to post-fungal invasion, because the majority of bats roosted in warm microsites where mortality rates were high, whereas a small proportion of the population roosted in cooler microsites with higher recapture rates.

~1 °C from pre-invasion to post-invasion, which represents a significant change for hibernating bat roosting conditions. After accounting for the regional decrease in roosting temperatures, there was no longer any relationship between bat recapture rates and invasion year, suggesting that selection primarily acted on pre-existing variability in bat microclimate use[34]. Although we used recapture rates as a proxy for bat survival (Fig. 4), some bats may not have been recaptured due to emigration or missed detections, but these effects would also need to be correlated with early hibernation temperatures to account for the relationships observed here. Conserving thermal refugia with relatively high survival or reproduction may facilitate persistence until host populations can evolve or until effective interventions can be developed, highlighting an important future management strategy for WNS and many other emerging infectious diseases[11,21,43,44].

However, despite the strong selection pressures imposed by temperature-dependent disease outcomes, most little brown bats in areas where WNS has been present for more than two years appear to be ecologically trapped in warm roosts. Our study design evaluated bat microclimate selection using a natural experiment: we compared bat roosting distributions before invasion, when population sizes were large and bats roosted across a wide range of temperatures, to bat distributions during and after pathogen invasion, when dramatic bat population declines left nearly all roosts in all hibernacula available for the few remaining bats to choose from. We found that in post-invasion years, most individual bats (52%) continued to hibernate in relatively warm microsites (>8 °C; Fig. 3b), and hibernaculum-level immigration and recruitment between March and November were highest for warmer hibernacula. Bats continue to choose these warm roosts even though the distances between the warm

and cold locations in our study were relatively short (e.g., 30 km between two hibernacula with 8.3 °C and 5.2 °C average temperatures). Therefore, warm roosts appear to be acting as preferred habitats, which might be threatening population persistence by attracting bats away from thermal refugia. Though warm temperatures have been identified as problematic for bats since WNS first emerged in the United States[21,24,37,45], it was not possible to identify warm roosts as ecological traps until individual-level data could be used to compare bat temperature preferences and bat recapture rates. In high priority areas for conservation, detailed mapping of all available early hibernation microclimates versus the microclimates that bats select could help to quantify exactly how strong bat preferences are for these high mortality roosts or whether preferences change with selection in the future. For example, one study from outside our region observed bats moving from relatively warm sections to relatively cold sections of some hibernacula after WNS was first detected[46]. Unless further evolution or microclimate modification interventions occur, the relative availability of warm, preferred habitats could determine local and global host population stability for the bats that are currently persisting with *P. destructans*.

Before *P. destructans* invaded, bats were already energetically constrained during hibernation; they had some optimal temperature range that minimized energy loss while balancing other physiological constraints, an upper thermal threshold that was too warm to allow them to hibernate, and a lower thermal threshold below which they would either drastically increase metabolic rates and energy expenditure or freeze[32–35]. Before invasion in Michigan and Wisconsin in particular, early hibernation roosting temperatures for little brown bats varied from 2.5 to 13.6 °C. During and after invasion, changes in fungal loads and

mortality were highest for bats in warm early hibernation roosts (i.e., >8 °C), where the majority of individual bats chose to roost. Therefore, our data suggests that relatively cool roosts might be "undervalued resources"—defined as unpreferred habitats with relatively high survival or reproduction[47]—that are used by relatively few bats in the post-invasion metapopulation. If bats were to use colder roosts more often, it might improve bat population persistence. However, cold roosts might only be advantageous in relatively warm years, because bats in cold roosts could be more susceptible to freezing during exceptionally cold winters[32,33]. Given this freezing risk, bats might be extremely thermally constrained post-invasion; there is a fine line between roosts that are too cold to survive unpredictable cold years and too warm to survive WNS.

The existence of traps, sinks, and refugia suggests several possible management options and future research needs to conserve host populations threatened by infectious diseases. To identify these habitats in the first place, longitudinal studies must be conducted in addition to the typical cross-sectional population surveys. In particular, by tracking individual host demographic parameters, post-invasion habitat suitability can be disentangled from host preferences for these habitats, allowing ecological traps to be identified when they might otherwise appear to be stable due to immigration from source habitats[48]. Understanding immigration and working with a metapopulation perspective will allow traps, sinks, and refugia to receive the most appropriate conservation interventions. Refugia for hosts can be created or conserved, which is already the norm in most species conservation programs and is already being considered to conserve amphibians threatened by the chytrid fungus[49,50]. In contrast, traps can be managed by deterring hosts from accessing those habitats (i.e., changing preferences to match suitability) or by modifying or treating the habitats to make them more suitable for survival/reproduction (i.e., changing suitability to match preferences)[47]. For WNS, the geomorphology of man-made warm hibernacula, like mines and tunnels, could be altered to slightly reduce average temperatures[32,37,46]. This type of environmental modification is commonly used to control infectious diseases that affect humans and domesticated species, but is rarely used as a wildlife conservation solution[11]. Together, these temperature-dependent strategies can reduce host population declines and promote long-term, regional stability of impacted populations.

## Methods

**Study design.** This study began in 2013, before *P. destructans* invaded Michigan and Wisconsin, and continued through 2020, after all Michigan and Wisconsin hibernacula were invaded by the pathogen. Throughout this period, we visited 22 hibernacula twice per winter during bat hibernation to quantify bat colony sizes, individual bat roosting temperatures, fungal loads (to which we added a constant 0.0001 before transforming to the $\log_{10}$ scale), and recapture probabilities (Fig. 2). We sampled bats during early hibernation in November, when more than 95% of swarm activity was expected to be finished[51], and again during late hibernation in March, when less than 1% of spring emergence activity was expected to have begun[51,52]. For each sampling event, we counted all bats of all species within the site. We focused our analyses on the little brown bat (*Myotis lucifugus*). This species suffered severe declines due to WNS[24,53], and was the only species abundant enough to provide sufficient sample sizes for our analyses. We divided bat counts by sections within the hibernaculum (i.e., "rooms") that potentially varied in microclimate, and we used HOBO Pro v2 data loggers to continuously record temperatures every 3 h in these sections (Supplementary Figs. 2 and 7).

After counting all bats, we haphazardly sampled 20–25 individual little brown bats stratified across sections and roughly in proportion to the number of bats in each section (Fig. 2). For each sampled bat, we used a Fluke 62 MAX IR laser thermometer to quantify the temperature of the substrate directly adjacent to the bat (<1 cm; the "roosting temperature"). We then used a previously developed protocol to collect a standardized swab of each bat's forearm and muzzle, which we stored in RNAlater until we could quantify fungal loads using qPCR[54]. Finally, we banded all swabbed bats that could be safely removed from their roosts (mean = 73%, median = 100% of all swabbed bats) with an aluminum lipped-band (2.9 mm;

Porzana Ltd., Icklesham, E. Sussex, U.K.), so that they could be re-sighted and individually-identified during subsequent visits (Fig. 2).

All sites and sampling procedures were covered by the appropriate state and federal permits, Virginia Tech IACUC protocol #17-180, and University of California, Santa Cruz IACUC protocol Kilpm1705. We followed field hygiene and decontamination protocols in accordance with United States Fish and Wildlife Service.

**Over winter changes in fungal loads on bats.** For infected, banded bats that were sampled during early hibernation and recaptured during late hibernation, we compared estimated fungal loads between the two time points to quantify how changes in fungal loads were affected by early hibernation roosting temperatures. In particular, we calculated fungal load change rates ($\lambda$) on individual bats using our estimates of fungal loads during early hibernation ($N_0$), fungal loads during late hibernation ($N_t$), and the number of weeks between collection of the early and late samples ($t$), where $\lambda = \ln((N_t/N_0)^{\wedge}(1/t))$. We excluded bats that were uninfected in November because if they had become infected between November and March, we could not have determined how long they had been infected ($t$ in the $\lambda$ equation) (Fig. 2).

Though the relationship between changes in fungal loads and early roosting temperatures on recaptured bats was linear over most of the range of early roosting temperatures that we observed for banded bats, fungal growth rates should be hump-shaped over large temperature ranges, with a minimum threshold temperature, an optimum temperature ($T_{opt}$), and a maximum threshold temperature ($T_{max}$). One such nonlinear temperature-dependent growth model is the Logan-10 function[55]:

$$\lambda(T) = a\left(\frac{1}{1+ce^{-pT}} - e^{-\frac{T_{max}-T}{\Delta T}}\right) \quad (1)$$

where $T$ is temperature, $\Delta T$ is the width of the upper boundary layer ($T_{max} - T_{opt}$), and $a$, $p$, and $c$ are shape parameters (Eq. 1). We fit this curve to our field-collected fungal load data.

This fungal load analysis did not include bats that were not recaptured, and thus the best-fitting temperature-dependent growth curve based on recaptured bats might be shallower than the curve for the whole population. We accounted for the potential missing bat bias using two methods that addressed the limited data from bats that roosted at relatively warm temperatures and bats that began hibernation with relatively high fungal loads. First, we expected that we would recapture relatively few bats that roosted near the optimum temperature for fungal growth, which could lead to model convergence issues. Therefore, we fit the temperature-dependent fungal growth curve in a Bayesian framework, using uniform priors for all parameters (except the maximum temperature for fungal growth, $T_{max}$) that were weakly informed by data from a previously-published laboratory fungus growth experiment (see below)[17]. Second, we expected that we would recapture relatively few bats that had high fungal loads during early hibernation, and that any recaptured bats with high early loads would have had lower changes in fungal loads that allowed them to survive their initially high loads. Therefore, we allowed the intercept of the temperature-dependent fungal growth curve to vary with early fungal load, where the curve for the lowest early fungal loads would be the least likely to be affected by disease-induced mortality.

We acquired reasonable ranges for $T_{opt}$, $T_{max}$, and the shape parameters using previously published fungal growth data from a laboratory experiment[17], where *P. destructans* was grown on Sabouraud dextrose agar plates for five weeks at each of nine average growth chamber temperatures: 0.8, 2.0, 4.6, 7.2, 11.9, 16.0, 17.6, 19.0, and 21.4 °C. The 7.2 °C treatment was repeated in three separate trials, which we combined for our analyses. We assumed that all temperatures had the same fungal carrying capacity ($K$) and selected a carrying capacity that minimized the mean squared error of model fits across all temperatures: $K = 0.85$ cm². Using this value and assuming that starting colony sizes ($N_0$) were the sizes observed at Week 1 during the laboratory experiment, we fit the logistic growth model ($K/(1 + ((K/N_0) - 1)*\exp(-r*Day))$) to the fungal colony size data from Weeks 2–5 (28 days of growth) to estimate intrinsic fungal growth rates ($r$) at each temperature. We then fit the temperature-dependent growth curve (Eq. 1) to the nine $r$ estimates using uninformative uniform priors for $a$, $p$, and $c$ and less flexible normal priors for $T_{opt}$ (mean = 14, SD = 5) and $T_{max}$ (mean = 21, SD = 1) in a Bayesian framework, which provided us with point estimates and 95% credible intervals for $a$, $p$, $c$, $T_{opt}$, and $T_{max}$. These models were run using three MCMC chains of 60000 iterations each, with a burn in of 30000 iterations. Model convergence was assessed using visual inspection of MCMC runs and by confirming that $\hat{R}$ values were less than 1.01. All analyses were run with package 'R2jags' in R[56].

We used the lab-derived 95% credible intervals as uniform priors for all parameters (except *Tmax*) in a second Bayesian analysis that fit the Logan-10 curve to the change in fungal load data from recaptured bats using three MCMC chains of 160000 (burn in = 80000). We also used an uninformative uniform prior for the load-dependent intercept term [$b$~unif(−0.5,0.5)]. Instead of estimating the thermal maximum, we set $T_{max}$ in the Logan-10 curve for the field data to a constant (the estimated thermal maximum from the laboratory study), because no bats roosted close to $T_{max}$. To confirm that these priors only weakly informed the model and that any reasonable curve shape would be possible, we ran a simulation to visualize the curve shapes that would be considered during the model fitting

process. In particular, we plotted Logan-10 growth curves for all combinations of the minimum, maximum, and median of the ranges for each parameter (Supplementary Fig. 1).

We used early roosting temperatures in our model because we expected that large changes in fungal loads during early hibernation would be more detrimental to bat survival than large changes in fungal loads during late hibernation[57]. We calculated how much variability in fungal load change ($\log_{10} \lambda$) the best-fitting load-dependent Logan-10 curve explained ($R^2$) by running a linear regression of observed versus predicted values. We also confirmed that there were significant correlations between $\log_{10} \lambda$ and early roosting temperatures and between $\log_{10} \lambda$ and early fungal loads using Spearman's rank correlation tests, which were performed in a frequentist framework without using priors from the laboratory experiment. We then repeated all analyses with late roosting temperatures instead of early temperatures and confirmed that late roosting temperatures always explained less variation (i.e., lower $R^2$), if any, in $\log_{10} \lambda$ than early roosting temperatures.

**Bat recapture rates**. Bats experiencing WNS symptoms often leave their hibernacula, where they are likely to die from exposure or starvation on the landscape[58]. In our study, these would be bats banded during early hibernation that left the hibernaculum before our March survey, which occurred weeks before average spring emergence dates[51]. Factors other than disease-induced mortality could have resulted in a failure to recapture bats (e.g., emigration/immigration, handling disturbance, non-disease related mortality), which would have reduced the likelihood of finding a significant effect of our predictors. However, only 5% of bats that were banded/sighted during early hibernation and missed during our late hibernation surveys were later re-sighted in a different year, and always in the same hibernaculum. Our model yielded qualitatively identical results whether we included these bats are recaptured or not in a given year. Some bats might also have died after our late hibernation survey, but before spring emergence, which would cause us to underestimate overall mortality. Therefore, we assume that recapture success was a conservative proxy for overwinter survival[22,59,60].

We used a mixed effects logistic regression with a logit link to quantify how early hibernation roosting temperatures and fungal loads affected the probability that bats infected during early hibernation were (=1) or were not (=0) re-sighted in late hibernation the same year. To quantify the predictive capabilities of this model, we performed 5-fold cross-validation on 1000 random divisions of our dataset, and calculated the average Area Under the Curve (AUC) for the resulting 1000 Receiver Operating Characteristic (ROC) curves, which can be interpreted as the percent of the test data that was successfully predicted by the training models.

Across invasion years, there was a trend towards increased recapture probabilities with time since WNS arrival (Fig. 5a), but including this covariate in the logistic recapture model with early hibernation temperatures and fungal loads resulted in very high variance inflation factors (VIF = 33.5). To determine whether invasion year explained any additional variation in recapture success, as might be expected if bats were evolving physiological resistance or tolerance to *P. destructans*, we ran a partial regression comparing the Pearson residuals for recapture success given early hibernation temperatures and fungal loads (logistic GLMM described above) to the Pearson residuals for invasion year given early hibernation temperatures and fungal loads (Poisson regression).

**Bat distribution shifts**. By 2020, there were 12 hibernacula that had harbored *P. destructans* for at least three years (>2 years is defined here as the post-invasion period) and that we had sampled during both the pre-invasion period (up to and including year 0, when WNS was first detected) and invasion period (years 1–2 of invasion). From pre-invasion to post-invasion, bat temperature distributions might have changed across these 12 hibernacula via two mechanisms: (1) bat preferences could have remained the same while populations declined in hibernacula and/or microsites with unfavorable temperatures, causing distribution shifts via selection; and/or (2) bat preferences could have changed towards avoidance of hibernacula and/or microsites with high temperature-mediated disease mortality, causing distribution shifts via selection or learning. In both cases, we would expect the average regional bat roosting temperatures to change over time. In contrast, we would only expect the average bat roosting temperature in any given site (or section) to change over time if the site provided a wide range of temperatures to select from, which was not always the case in the hibernacula we studied. Therefore, we quantified how the regional mean early hibernation roosting temperatures for all counted bats changed across all 12 of these hibernacula from pre-invasion to post-invasion (2013–2020, up to four years after the fungus was first detected) using a Gaussian regression, without including a random site effect. We visualized this change using density plots with 1 °C smoothing bin widths.

Though we stratified bat roosting temperature sampling by section, roughly in proportion to the total number of bats present in a section, we only sampled up to 25 little brown bats per site, and we could not sample exactly in proportion to the total bats present. Therefore, calculating the mean temperatures based on only sampled bats might be misleading. Instead, we used the observed temperatures from our sampled bats to estimate the roosting temperatures for bats that were counted but not sampled. In particular, we randomly generated an additional set of temperatures for all unmeasured, counted bats based on the mean and standard deviation of the roosting temperatures of sampled bats in the same section during

the same survey. We then used both known (sampled bats) and estimated (counted bats) temperatures in our Gaussian regression to determine how average temperatures changed from pre-invasion to post-invasion. We also confirmed that qualitative results were the same if we only used the temperatures from sampled bats, without estimation. By considering how bats' early hibernation temperature distributions changed throughout invasion, regardless of infection status, we were quantifying bat roosting preferences within and between hibernacula, rather than behavioral responses to infection within a season (i.e., "behavioral fever").

Finally, we used the number of bats counted per hibernaculum during early hibernation (November) and late hibernation (March) in all hibernacula that still had bats in post-invasion years to calculate two demographic parameters: over summer population growth rates (reflecting immigration and recruitment to sites between March and November) and over winter mortality rates (population growth rates from November to March). We used a Gaussian linear model to determine whether the $\log_{10}$ population growth rates were correlated with average early hibernation roosting temperatures for sampled bats in each hibernaculum, the demographic season (over summer versus over winter), and the interaction between temperature and demographic season. A significant interaction term would indicate that temperature-correlated bat immigration and recruitment rates were decoupled from temperature-mediated population declines.

All analyses described above were performed in R version 3.5.1[61] with packages 'ggplot2'[62], 'lme4'[63], 'loo'[64], 'pROC'[65], and 'R2jags[56]. Visual inspection of residuals and predictions plots confirmed acceptable model fits for all regression models.

**Reporting summary**. Further information on research design is available in the Nature Research Reporting Summary linked to this article.

## Data availability
The datasets generated during this study are available in the VTechData repository [https://doi.org/10.7294/7134-4610][66].

## Code availability
All R scripts used in our analyses are available in the VTechData repository [https://doi.org/10.7294/7134-4610][66].

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

## Acknowledgements

We thank Steffany Yamada, Ricardo Acevedo, Rowan Berman, Mercedes Clark, Cyrus Crevits, Molly Kwitny, and Kiarra Parthasarathy for assistance in data management and curation and Jeff Foster and Katy Parise for testing fungal swabs. We also thank the numerous land owners and site managers that provided continued access to the sites. Funding was provided by the joint NSF-NIH-NIFA Ecology and Evolution of Infectious Disease award DEB 1911853, DEB-1336290, and DEB-1115895, and the USFWS (F17AP00591).

## Author contributions

Conceptualization: S.R.H., J.R.H., and K.E.L.; Methodology: S.R.H., J.R.H., and K.E.L.; Formal analysis: S.R.H., J.R.H., and K.E.L.; Investigation: J.E.D., J.R.H., A.M.K., H.M.K., K.E.L., J.A.R., W.H.S., and J.P.W.; Data curation: S.R.H., J.R.H., and K.E.L.; Writing–original draft: S.R.H.; Writing–review and editing: J.E.D., S.R.H., J.R.H., A.M.K., H.M.K., K.E.L., J.A.R., W.H.S., and J.P.W.; Visualization: S.R.H.; Supervision: J.R.H. and K.E.L.; and Project administration: J.R.H., A.M.K., and K.E.L.

## Competing interests

The authors declare no competing interests.
