## [Peer Review File · Nature Communications]

Reviewers' comments:

Reviewer #1 (Remarks to the Author):

Hopkins et al present a novel and important study testing the role of ecological traps play in an emerging infectious disease. Impressively, authors monitored thousands of bat hosts before, during, and after invasion of white-nose syndrome. Interestingly, they found that individuals hibernating in relatively warm microhabitats were less likely to survive than those hibernating in relatively cool microhabitats. This temperature dependent survival seems to have resulted in a selection event that slightly decreased the average roosting temperature. However, the majority of bats continue to prefer relatively warm microhabitats. Thus, the authors suggest that these warm microhabitats are ecological traps. I really enjoyed reading and reviewing this manuscript.

I had two main concerns with the manuscript that should be addressed.

1) There are improvements that could be made in the introduction and discussion which would strengthen the author's arguments and broaden the impact of this paper. I think the review of relevant literature is a little lacking. Many of the ideas in this paper have been heavily explored in the amphibian chytridiomycosis system. Including some of those examples would strengthen the arguments of this study. I was actually pretty surprised by the exclusion of chytrid literature in the introduction and discussion, given the similarities between the two systems. Variation in chytrid susceptibility driven by the microhabitat temperature selected by hosts has been tested in the field and the lab (e.g. Rowley & Alford 2013 Sci Reports, Forrest & Schlaepfer 2011 PloS ONE, Sauer et al 2018 Proc B, Richards-Zawacki 2010 Proc B). There is also a lot of research on altering habitat to reduce chytrid disease risk, including research on altering microhabitat temperatures. Garner et al (2016) wrote good review of the numerous habitat manipulations that had been proposed and tested in this system (<https://doi.org/10.1098/rstb.2016.0207>). Habitat manipulation is probably the main management strategy being pursued right now in places where the impacts of chytridiomycosis invasion have been particularly devastating. There has been substantial discussion of the costs and benefits to altering the environment to mitigate disease (including altering temperatures), much of which would apply to the management strategies proposed in this study.

2) The justification for using in vitro WNS growth rates as a prior for modeling in vivo growth is completely absent. It cannot be assumed that pathogen thermal performance in culture is predictive of growth on hosts. Disease is an interaction between a host and pathogen, meaning both the host and pathogen's thermal performances impact the outcome of that interaction. This is especially true in systems where the host is ectothermic, which the case with hibernating bats (Cohen et al 2016 Ecology Letters is probably the best example of this idea).

Minor comments:

Abstract: The second sentence seems like an abrupt transition from the first sentence.

Line 116: "hibernation (20-23), and over.." It is unclear what these numbers are referring to, there is no unit and no obvious context for them.

Line 146: There is a missing a space between (<1 cm) and (i.e. the roosting temperature).

Line 149-159: What exactly does "we banded as many swabbed bats as possible with an aluminum band" mean? It would be helpful to have more information on the % of swabbed bats banded at each section and whether or not that % is more or less equal across sections and hibernaculum.

Line 171-173 & 224-225: See main comment #2.

I think the discussion of results would be clearer if more explicit parallels were made between the results and the hypotheses laid out in the conceptual figure from the introduction. Why is Figure 1 post-invasion panel B happening and not post-invasion panel A?

Line 236: "For instance, for WNS..." the double 'for' is a bit awkward.

Figures: Text in figures 2-5 could really use a font increase. The numbers on the axis are very difficult to read.

Reviewer #2 (Remarks to the Author):

Dear Hopkins and Colleagues,

I appreciated the opportunity to review your manuscript on ecological traps and thermal refugia in the context of white-nose syndrome (WNS). This is an important topic for conservation biologists, and your collaboration here provides data that will be useful to the community. The main findings, that fungal loads increased more on bats initially captured at warm sites, that recapture rates were lower for bats initially captured at warmer locations, and that a shift in habitat use is occurring post-WNS fit nicely with the existing literature. Thus, the results tell a cohesive story that are mostly well supported with data.

However, the manuscript doesn't acknowledge the existing literature that connects to this study. Much of your findings (although so very important) are confirming previous studies. But you decided not to note that other field and laboratory studies have shown temperature dependent trends in survival of bats infected with *P. destructans*. Or that other field studies have shown bats moving into colder hibernacula in the years after WNS colonizes an area. Or that modeling studies predicted these results and warned of the presence ecological traps in comparison to cold refugia. Finally, although others have written about managing hibernacula temperatures in response to WNS and previous conservation threats, here, too the current work is presented on an island of its own. So while this is very important and well-conducted study, it is presented with an exaggerated sense of novelty. I appreciate your concise writing and do not believe that a citation of all relevant works is needed. That would be unfair. However, the extent to which several researchers' work is left out of your paper gives me the sense that you are either unaware that many have papers seen and said the same things or that you are ignoring their work. Given that the context of your study is a conservation crisis, I respectfully recommend that you try more to put your work within that context. I have included several papers that are too similar to this work to be ignored. However, it's not intended to be comprehensive.

Additionally, I have several comments on the presentation of the manuscript and the methods used. I have organized these below in terms of style and substance.

Style

- The manuscript seems to retain some formatting from a previous submission. For example, there seem to be some references cited using numerical in-text citations while the majority are written out as author and year (for example, Line 116).
- Forgive me if I am wrong, but I believe you introduce abbreviations such a T_{max} without stating what those are, or in this case, how they are measured (Line 172).
- You occasionally write long, compound sentences where shorter statements would be easier to understand. For example, you start the Methods section with the following:
"Beginning in 2013, before *P. destructans* invaded Michigan and Wisconsin, and continuing through present (2019), when almost all Michigan and Wisconsin hibernacula have been invaded, we visited 22 hibernacula twice per year during bat hibernation (October to April) to quantify bat colony sizes and individual bat roosting temperatures, fungal loads (to which we added a constant

0.0001 before transforming to the log₁₀ scale), and recapture probabilities."

- Overall, I suggest careful reviewing this manuscript for these types of errors and places where sentence structure can be improved for clarity.

Substance

- In the introduction you write that, "Individual bats can select microclimates that optimize temperature-dependent energy use with other physiological constraints during hibernation". I am not sure the references you intended to cite because they are cited numerically, but I have included two more recent works by Boyles and colleagues that would be more updated than what I saw in the lit cited (see references at end).

- In Line 87 and throughout you discuss ecological traps in the context of hibernacula temperature. This has been written about and studied before. In a paper from 2018, Lilley and colleagues predicted exactly this, and concluded their article with the following: "Our model suggests that prioritizing the preservation or restoration of high-quality habitat, which were optimal for *M. lucifugus* hibernation prior to WNS, may enhance the spread and impact of Pd. In areas where Pd has become established and endemic, these hibernacula with suitable resources/climate may become metapopulation-scale management traps: sites that are normally beneficial and attractive to bats, but become a net drain on the metapopulation owing to the impacts of disease". This paper, published in 2018, cannot be left out of your manuscript as it establishes many of the ideas you pursue here.

- At Line 112 you discuss how different species have unique preferences for winter microclimates. Why, then, only present data for the little brown bat? There is far more data available for the little brown bat than is available for other species, why discard valuable information? It seems odd to expect that if species would have different preferences pre-WNS that they would all respond to the disease the same way. Seems like a missed opportunity here to think beyond a single species approach. More importantly, there are data showing that some of these species have changed their winter habitat use post-WNS. I don't see how you can present a study on the effect of temperature on bat distribution shifts without citing works on the exact same topic.

- Lines 133-137. You write, "we visited 22 hibernacula twice per year during bat hibernation (October to April) to quantify bat colony sizes and individual bat roosting temperatures, fungal loads (to which we added a constant 0.0001 before transforming to the log₁₀ scale), and recapture probabilities. For each sampling event during early (November) and late (March) hibernation, we counted all bats of all species within the site." There is a small but possibly important discrepancy here. The beginning of this quote implies that early hibernation data were collected in October while the end implies November. Similarly, were late winter data collected in March or April?

- On the matter of when data were collected, you later make a key assumption that bats that were not recaptured died. As described below, I don't think this is necessarily a sound assumption. In addition to matters described below, when one visits the hibernaculum is an important factor in predicting animal re-encounters. As described in the study from Manitoba, entrance and emergence from hibernation occurs over several months with males and females exhibiting unique timings. Thus, the timing of data collection is critical to clarify. Furthermore, that some bats may have left their hibernacula, regardless of in late winter visit was in March or April, must be acknowledged.

- Line 148: perhaps you are also looking at gene expression in a different aspect of this study, nullifying this comment, but you may be able to make your lab extractions easier if you forgo the RNA later. Or at least save some funds.

- Line 154: you may want to check this IACUC reference. From the outside looking in, it seems odd to have a protocol under the name of someone not involved in this study. Of course, there could be many explanations. Simply a thought to double check.

- Line 156 and throughout: you did not measure fungal growth. You measured the amount of fungal DNA collected with a swab of the same bats at two points in time. This isn't growth, and presenting it as such could be confusing. The bats can spread the fungus among themselves and hibernacula serve as reservoirs. Because bats are constantly moving about hibernacula in winter and coming into contact with each other and new areas of the hibernaculum, calling changes in

loads growth is not well defended.

- Lines 184-185. The assumption that not re-encountering a bat in late winter means the animal has died needs to be supported. As described above, bats vary in their timing of arrival at and departure from the hibernacula, and this activity might include time points after the first survey and before the last (see citation below). I think your assumption needs to be put in the context of this knowledge. Furthermore, bats do move among hibernacula during winter. This is a very poorly studied phenomenon for obvious reasons, but early studies clearly show this occurs. And more recent studies show bats are seen in different hibernacula among years, without clear indication of when the movement occurred. Finally, bats move about hibernacula and the efficacy of surveys is not 100%. You actually show this in your supplemental methods where you mention finding bats in later years that were previously missed. Thus, I think this assumption is far over-stated as being "likely". An exception to this would be during the mass mortality period associated with WNS arrival, but I would not apply that to the years earlier or later. This does not mean that your data can't be analyzed similar to how they currently are. However, I think that similar to fungal growth rates, these data need to be presented as representing what they actual are. Recapture rates.

- Line 214. Eliminate this section. There is so much more about your analyses in other sections that this seems odd to me.

- Discussion. The only WNS paper you discuss here are from the Langwig lab. I do not object to self-citations, that would be unfair and hypocritical, but others have found similar results as you, and yet those studies are simply ignored. This limits the contribution of your work because it places it in a vacuum. For example, you cite Langwig 2016 to support the idea of management of hibernacula temperatures. Yet, that 2016 paper only discusses that idea, and does not present data on the matter. Those data and studies do exist (see Johnson et al 2016 and Richter et al 1993 below).

In close, I have tremendous respect and appreciation for the valuable data presented in this manuscript. But I also think the manuscript could be tremendously improved by putting the study into context and considering the work that others have done. How are you results similar? How are they unique? This is more useful for the larger community. The WNS landscape in North America is large and different regions will have unique stories and species. All that is missed when we fail to recognize the contributions of others. Equally important, I do think the findings need to be presented in a more balanced way (recaptures are not survival, fungal loads are not fungal growth). In this way the contribution will be more readily apparent and have greater impact on the field.

References

Boyles, J. G., Boyles, E., Dunlap, R. K., Johnson, S. A., & Brack, V. (2017). Long-term microclimate measurements add further evidence that there is no "optimal" temperature for bat hibernation. *Mammalian Biology*, 86(1), 9-16.

Boyles, J. G., Johnson, J. S., Blomberg, A., & Lilley, T. M. (2020). Optimal hibernation theory. *Mammal Review*, 50(1), 91-100.

Grieneisen, L. E., Brownlee-Bouboulis, S. A., Johnson, J. S., & Reeder, D. M. (2015). Sex and hibernaculum temperature predict survivorship in white-nose syndrome affected little brown myotis (*Myotis lucifugus*). *Royal Society open science*, 2(2), 140470.

Humphrey, S. R., & Cope, J. B. (1976). Population ecology of the little brown bat, *Myotis Lucifugus*, in Indiana and north-central Kentucky. *American Society of Mammalogists*.

Johnson, J. S., Reeder, D. M., McMichael III, J. W., Meierhofer, M. B., Stern, D. W., Lumadue, S. S., ... & Kath, J. A. (2014). Host, pathogen, and environmental characteristics predict white-nose syndrome mortality in captive little brown myotis (*Myotis lucifugus*). *PLoS One*, 9(11).

Johnson, J. S., Scafani, M. R., Sewall, B. J., & Turner, G. G. (2016). Hibernating bat species in Pennsylvania use colder winter habitats following the arrival of white-nose syndrome. *Conservation and Ecology of Pennsylvania's Bats*. Bradford: The Pennsylvania Academy of Sciences, 181-199.

Lilley, Thomas Mikael, et al. "White-nose syndrome survivors do not exhibit frequent arousals associated with *Pseudogymnoascus destructans* infection." *Frontiers in zoology* 13.1 (2016): 12.

Lilley, T. M., Anttila, J., & Ruokolainen, L. (2018). Landscape structure and ecology influence the spread of a bat fungal disease. *Functional ecology*, 32(11), 2483-2496.

Norquay, K. J. O., & Willis, C. K. R. (2014). Hibernation phenology of *Myotis lucifugus*. *Journal of Zoology*, 294(2), 85-92.

Richter, A. R., Humphrey, S. R., Cope, J. B., & Brack Jr, V. (1993). Modified cave entrances: thermal effect on body mass and resulting decline of endangered Indiana bats (*Myotis sodalis*). *Conservation Biology*, 7(2), 407-415.

Reviewer #3 (Remarks to the Author):

This very interesting manuscript synthesizes an impressive set of field data relating to bats hibernating in sites while parasitized by the fungus that causes white-nose syndrome. The introduction section sets up the story in the context of ecological traps and I was quite frankly excited to see the evidence and conclusions by the end of the introduction (despite its shallow depth of literature review regarding bat hibernation and WNS beyond the author line). Try as I might, after reading through the methods, results, discussion sections, I was not convinced the promise implied by the title and intro was fulfilled. Assumptions in an ecological inference study of this nature are everything, and I wasn't convinced that the assumptions of this study were met. First, no convincing evidence is presented or cited to clearly demonstrate that the quantity of fungal gene fragments present on the swabbed wings of bats at any given time is directly and closely associated with disease severity in that individual. Fungal particles on the surfaces of gregarious animals in roosts where the pathogen exists in the environment too seems unlikely to be closely coupled as an index of individual health. Second, the assumption that bats not found after banding was a good proxy for overwinter survival seems like a stretch, because bats can abandon sites due to banding disturbance or simply move elsewhere. Furthermore, banding was discontinued in North America for several decades due to known survival effects, so it isn't possible to rule out the possibility that it was the banding that altered the very overwinter survival being measured. Third, all of these measures seem to assume that the bats stayed put in the hibernation sites throughout winter and that the temperatures measured by the data loggers were those the bats actually experienced. Individual bats can move during winter and where you find them during period checks may not reflect the temperature conditions they were seeking before or after that snapshot (single sampling period) of time. Fourth, temperature is the only microclimate variable considered, even though prior work with WNS and hibernating bats built a large body of evidence (empirical and theoretical) indicating temperature interacts dynamically with humidity to influence survival. There also seems to be little consideration of seasonally dynamic changes in temperature within the hibernation sites and the options individual bats have to move among sites during winter to fit their thermoregulatory needs.

Maybe I missed something, but the metrics used to build the case for bats funneling into ecological trap conditions during winter just don't seem suited for the rock-solid kind of evidence needed to support such a claim.

Reviewers' comments:

Reviewer #1 (Remarks to the Author):

Hopkins et al present a novel and important study testing the role of ecological traps play in an emerging infectious disease. Impressively, authors monitored thousands of bat hosts before, during, and after invasion of white-nose syndrome. Interestingly, they found that individuals hibernating in relatively warm microhabitats were less likely to survive than those hibernating in relatively cool microhabitats. This temperature dependent survival seems to have resulted in a selection event that slightly decreased the average roosting temperature. However, the majority of bats continue to prefer relatively warm microhabitats. Thus, the authors suggest that these warm microhabitats are ecological traps. I really enjoyed reading and reviewing this manuscript.

Thank you! We appreciate your feedback. We realized while reading this comment that the average change in roosting temperatures might seem small without reference, but a 1°C average shift is actually large for hibernating bats. We have clarified this in the manuscript.

I had two main concerns with the manuscript that should be addressed.

1) There are improvements that could be made in the introduction and discussion which would strengthen the author's arguments and broaden the impact of this paper. I think the review of relevant literature is a little lacking. Many of the ideas in this paper have been heavily explored in the amphibian chytridiomycosis system. Including some of those examples would strengthen the arguments of this study. I was actually pretty surprised by the exclusion of chytrid literature in the introduction and discussion, given the similarities between the two systems. Variation in chytrid susceptibility driven by the microhabitat temperature selected by hosts has been tested in the field and the lab (e.g. Rowley & Alford 2013 Sci Reports, Forrest & Schlaepfer 2011 PloS ONE, Sauer et al 2018 Proc B, Richards-Zawacki 2010 Proc B). There is also a lot of research on altering habitat to reduce chytrid disease risk, including research on altering microhabitat temperatures. Garner et al (2016) wrote good review of the numerous habitat manipulations that had been proposed and tested in this system (<https://doi.org/10.1098/rstb.2016.0207>). Habitat manipulation is probably the main management strategy being pursued right now in places where the impacts of chytridiomycosis invasion have been particularly devastating. There has been substantial discussion of the costs and benefits to altering the environment to mitigate disease (including altering temperatures), much of which would apply to the management strategies proposed in this study.

We agree that WNS and chytrid have important parallels and appreciate the suggestion to discuss these parallels more. We also think that this comment suggests that we were not clear enough about how our work differs from work in other systems, so we have made several clarifying edits.

- For both WNS and chytrid, and many other disease systems, we agree that it is well-established that infection/susceptibility can be temperature dependent. This is a main point that we make in our introduction, and we have added the citations you suggested.
- We also agree that the chytrid system is a great example for thermal refugia: because frogs in warmer microclimates are more resistant to disease, many studies have recommended that conserving warm habitats or even altering microclimates might create thermal refugia for frogs. We have added citations to these relevant papers where we discuss the importance of conserving or creating refugia.
- The focus on thermal preferences in frogs has mostly involved studies on “behavioral fever”, where infected frogs might move into warmer microclimates when they are infected to try to clear their infections. Therefore, measurements of the temperature preferences of individual frogs have been made within an epidemic/season. In contrast, our study looks at bat preferences and distributions across many years from before to after pathogen invasion, where the metapopulation experiences yearly epidemics and mortality events. We have added a brief clarification that we were looking at long-term bat microclimate preferences in this study, rather than behavioral fever.
- Finally, and most importantly, we want to note that the main goal of our study was to determine whether *P. destructans* created ecological traps for bats, because ecological traps created by invasive pathogens have not previously been demonstrated using the appropriate individual-level data. We did not mean to imply that this was the first study to look at creating or conserving thermal refugia from disease. We have edited the text to better reflect this.

2) The justification for using in vitro WNS growth rates as a prior for modeling in vivo growth is completely absent. It cannot be assumed that pathogen thermal performance in culture is predictive of growth on hosts. Disease is an interaction between a host and pathogen, meaning both the host and pathogen’s thermal performances impact the outcome of that interaction. This is especially true in systems where the host is ectothermic, which the case with hibernating bats (Cohen et al 2016 Ecology Letters is probably the best example of this idea).

Thank you for pointing out that we needed to explain this better! We have clarified this in several ways in the manuscript:

- A correlation test demonstrates that it is not necessary to incorporate laboratory growth rates to show that changes in fungal loads are correlated with temperature in the wild. We have clarified this in the main text.
- We also went beyond a correlation analysis and fit a unimodal relationship to the fungal load change data, because we believe that is the most biologically appropriate analysis. There are several functions that can describe a unimodal temperature-dependent growth relationship, all of which are so flexible that if parameters are completely unconstrained, they can produce biologically implausible shapes (e.g.,

including negative numbers), or the models can fail to converge. Therefore, we constrained the parameters to reasonable values using the laboratory growth rates from a previous study. Importantly, we used uniform priors, which don't weight estimates towards a mean value, and the parameter space covered by the ranges on our uniform priors could have produced any reasonable unimodal curve shape within the range of our data, as we now show with a new supplemental simulation analysis and figure.

- Finally, we just want to note that this is one way that the WNS system is surprisingly different from the chytrid system. In the chytrid system, ectothermic hosts' immune responses depend on hosts' body temperatures. Though bats can mount some immune response to the fungus, hibernating animals also downregulate their immune systems. We therefore expect any temperature-dependent immune response to be small relative to how much fungal growth rates vary across temperatures (several orders of magnitude). However, as we describe above, our priors allowed for any reasonable curve shape, so if bat immune responses were somehow shifting the temperature-dependent growth curve, we still would be capturing that in our analysis.

Minor comments:

Abstract: The second sentence seems like an abrupt transition from the first sentence.

Thank you for pointing this out. We have changed these two sentences to flow better.

Line 116: "hibernation (20-23), and over.." It is unclear what these numbers are referring to, there is no unit and no obvious context for them.

It appears that our citation software had a problem during exporting or sharing, and some of the citations were saved in the wrong format. We have corrected this.

Line 146: There is a missing a space between (<1 cm) and (i.e. the roosting temperature).

We have corrected this.

Line 149-159: What exactly does "we banded as many swabbed bats as possible with an aluminum band" mean? It would be helpful to have more information on the % of swabbed bats banded at each section and whether or not that % is more or less equal across sections and hibernaculum.

We thank the reviewer for drawing our attention to this. To clarify, we can sometimes swab a bat but not band the bat, because swabbing can be done without removing the bat from the wall or ceiling, whereas banding requires having the bat in hand. Bats sometimes roost in cracks and crevices making them possible to swab but impossible to remove and band. We have clarified this and added the percent of swabbed bats that were also banded.

We also want to note that banded bats covered the full range of temperatures used by swabbed bats, as shown by the distributions in our Fig. 5.

Line 171-173 & 224-225: See main comment #2.

Thank you for pointing out these specific line numbers. As described above, we clarified our use of the laboratory growth rate priors.

I think the discussion of results would be clearer if more explicit parallels were made between the results and the hypotheses laid out in the conceptual figure from the introduction. Why is Figure 1 post-invasion panel B happening and not post-invasion panel A?

We agree with the reviewer and have now better drawn the parallel between this outcome and panel B in Fig. 1 in the discussion.

Line 236: “For instance, for WNS...” the double ‘for’ is a bit awkward.

We thank the reviewer for their careful reading. We removed the “For instance” transitional phrase.

Figures: Text in figures 2-5 could really use a font increase. The numbers on the axis are very difficult to read.

We have increased the font sizes for the axes in all of these figures.

Reviewer #2 (Remarks to the Author):

Dear Hopkins and Colleagues,

I appreciated the opportunity to review your manuscript on ecological traps and thermal refugia in the context of white-nose syndrome (WNS). This is an important topic for conservation biologists, and your collaboration here provides data that will be useful to the community. The main findings, that fungal loads increased more on bats initially captured at warm sites, that recapture rates were lower for bats initially captured at warmer locations, and that a shift in habitat use is occurring post-WNS fit nicely with the existing literature. Thus, the results tell a cohesive story that are mostly well supported with data.

Thank you!

However, the manuscript doesn't acknowledge the existing literature that connects to this study. Much of your findings (although so very important) are confirming previous studies. But you decided not to note that other field and laboratory studies have shown temperature dependent trends in survival of bats infected with *P. destructans*. Or that other field studies have shown bats

moving into colder hibernacula in the years after WNS colonizes an area. Or that modeling studies predicted these results and warned of the presence ecological traps in comparison to cold refugia. Finally, although others have written about managing hibernacula temperatures in response to WNS and previous conservation threats, here, too the current work is presented on an island of its own. So while this is very important and well-conducted study, it is presented with an exaggerated sense of novelty. I appreciate your concise writing and do not believe that a citation of all relevant works is needed. That would be unfair. However, the extent to which several researchers' work is left out of your paper gives me the sense that you are either unaware that many have papers seen and said the same things or that you are ignoring their work. Given that the context of your study is a conservation crisis, I respectfully recommend that you try more to put your work within that context. I have included several papers that are too similar to this work to be ignored. However, it's not intended to be comprehensive.

We thank the reviewer for their thoughtful reply and providing citations relevant to our work. We have addressed the bat and WNS specific citations covered in our paper in several ways in response to your comment:

- **Most importantly, we added many citations to our paper that support the aspects of bat and WNS ecology that readers need to understand our results. We predominantly added extra supporting citations to existing sentences where we had originally cited one or two key papers only, because our manuscript was originally submitted to a journal with a small reference limit. Nature Communications articles allow up to 70 references, so we have added many references during this revision.**
- **As noted above, some of our references became unlinked from our citation manager, so some of the citations that you mention were intended to be included in our original submission, but only showed up as numbers in parentheses. We apologize for this oversight and have fixed these issues.**
- **We agree that we are not the only scientists to recognize that temperature plays a critically important role in determining disease outcomes from WNS, nor are we the only people to hypothesize that warm hibernacula might be problematic for bats. However, there is an important distinction between hypotheses generated by modeling studies or population-level surveys and evidence demonstrating ecological traps. For example, previous surveys showing that bats within single hibernacula have shifted to use colder sections post-invasion do *not* demonstrate that *ecological traps* exist; they use population-level inferences to suggest that bats may be using *thermal refugia*. In contrast, our dataset tracks individual bats to determine their microclimate use, infection loads, and disease outcomes from pre- to post invasion, and shows that bats are continuing to prefer and use the warm sites where pathogen growth rates are highest and recapture rates are lowest, even when cooler refugia are available. This is the first study to demonstrate that ecological traps exist in this system, which is important for bat conservation. Furthermore, this is also the first study to use individual-level data to show that an invasive pathogen created ecological traps for hosts in *any* disease system, and thus we do not think that we are exaggerating the novelty of our work. The lack of individual-level data in ecology**

and evolution is frequently highlighted as a major limitation of the field (e.g. Dobson, Tilman & Holt, Unsolved Problems in Ecology, Tim Coulson's chapter: "Ecology and Evolution is hindered by the lack of individual-based data"); these data are necessary to make fundamental theoretical advances, including the identification of ecological traps. We have clarified these points in several places in our manuscript.

Additionally, I have several comments on the presentation of the manuscript and the methods used. I have organized these below in terms of style and substance.

Style

- The manuscript seems to retain some formatting from a previous submission. For example, there seem to be some references cited using numerical in-text citations while the majority are written out as author and year (for example, Line 116).

Thanks for pointing this out. Again, we apologize for the citation problems. It seems to be a citation software error that we have fixed.

- Forgive me if I am wrong, but I believe you introduce abbreviations such a Tmax without stating what those are, or in this case, how they are measured (Line 172).

Good point! We had defined everything in the supplement, but not the main text. We've corrected this throughout the manuscript.

- You occasionally write long, compound sentences where shorter statements would be easier to understand. For example, you start the Methods section with the following:

"Beginning in 2013, before *P. destructans* invaded Michigan and Wisconsin, and continuing through present (2019), when almost all Michigan and Wisconsin hibernacula have been invaded, we visited 22 hibernacula twice per year during bat hibernation (October to April) to quantify bat colony sizes and individual bat roosting temperatures, fungal loads (to which we added a constant 0.0001 before transforming to the log₁₀ scale), and recapture probabilities."

- Overall, I suggest careful reviewing this manuscript for these types of errors and places where sentence structure can be improved for clarity.

We changed this to better reflect our sampling schedule, as described in other places, and broke it into multiple sentences.

Substance

- In the introduction you write that, "Individual bats can select microclimates that optimize temperature-dependent energy use with other physiological constraints during hibernation". I am not sure the references you intended to cite because they are cited numerically, but I have included two more recent works by Boyles and colleagues that would be more updated than what I saw in the lit cited (see references at end).

See above; we attempted to cite several of the papers that you mention, but there was an error (now corrected) which dropped several of the citations from the literature cited.

- In Line 87 and throughout you discuss ecological traps in the context of hibernacula temperature. This has been written about and studied before. In a paper from 2018, Lilley and colleagues predicted exactly this, and concluded their article with the following: "Our model suggests that prioritizing the preservation or restoration of high-quality habitat, which were optimal for *M. lucifugus* hibernation prior to WNS, may enhance the spread and impact of Pd. In areas where Pd has become established and endemic, these hibernacula with suitable resources/climate may become metapopulation-scale management traps: sites that are normally beneficial and attractive to bats, but become a net drain on the metapopulation owing to the impacts of disease". This paper, published in 2018, cannot be left out of your manuscript as it establishes many of the ideas you pursue here.

We agree that Lilley 2018 provides important theoretical predictions on the effect of temperature on *P. destructans* spread. Our study did not address spread, because all our sites were invaded at roughly the same time, regardless of their temperatures. Rather, we quantified bat temperature preferences as *P. destructans* invaded and the effects of those individual-level choices on disease outcomes. Therefore, we want to note that we use the term thermal refugia to reflect areas where pathogen growth is low, not areas where the pathogen has not invaded (as it is used in Lilley 2018). We have clarified this in the text.

- At Line 112 you discuss how different species have unique preferences for winter microclimates. Why, then, only present data for the little brown bat? There is far more data available for the little brown bat than is available for other species, why discard valuable information? It seems odd to expect that if species would have different preferences pre-WNS that they would all respond to the disease the same way. Seems like a missed opportunity here to think beyond a single species approach. More importantly, there are data showing that some of these species have changed their winter habitat use post-WNS. I don't see how you can present a study on the effect of temperature on bat distribution shifts without citing works on the exact same topic.

We apologize for the miscommunication and thank the reviewer for drawing our attention to this. We did not discard valuable information regarding other bat species; there is simply not enough individual-level mark-recapture data for species other than the little brown bat from our study or any other study to address whether ecological traps exist for those other species. We have hopefully addressed any confusion about multiple species by clarifying our methods section.

- Lines 133-137. You write, "we visited 22 hibernacula twice per year during bat hibernation (October to April) to quantify bat colony sizes and individual bat roosting temperatures, fungal loads (to which we added a constant 0.0001 before transforming to the log₁₀ scale), and

recapture probabilities. For each sampling event during early (November) and late (March) hibernation, we counted all bats of all species within the site." There is a small but possibly important discrepancy here. The beginning of this quote implies that early hibernation data were collected in October while the end implies November. Similarly, were late winter data collected in March or April?

We thank the reviewer for drawing our attention to this confusing sentence. We meant to explain that bats hibernate from October to April, and that we sampled bats in November (early hibernation) and March (late hibernation). We have clarified this point in the manuscript.

- On the matter of when data were collected, you later make a key assumption that bats that were not recaptured died. As described below, I don't think this is necessarily a sound assumption. In addition to matters described below, when one visits the hibernaculum is an important factor in predicting animal re-encounters. As described in the study from Manitoba, entrance and emergence from hibernation occurs over several months with males and females exhibiting unique timings. Thus, the timing of data collection is critical to clarify. Furthermore, that some bats may have left their hibernacula, regardless of in late winter visit was in March or April, must be acknowledged.

We thank the reviewer for drawing our attention to this valuable point. We also address how emigration and immigration might have affected our results using a simulation, described in more detail below. We chose our visits schedule based on a prior study in our region which used beam break data to estimate hibernation onset and spring emergence. Specifically, using a 13 y beam break study, Meyer *et al* (2016) found that >95% of bats are in hibernation before November 1st and <1% of bats have left by mid-March. We have clarified how our sampling scheduled corresponded to bat life cycles in the text.

- Line 148: perhaps you are also looking at gene expression in a different aspect of this study, nullifying this comment, but you may be able to make you lab extractions easier if you forgo the RNAlater. Or at least save some funds.

Thank you for this suggestion. We have already sampled all of these bats and finished the extractions, and because recent comparisons have demonstrated that qPCR analysis of samples stored in water were less sensitive than paired samples stored in RNAlater (J. Evans, 2020 WNS Webinar), we will likely continue to use RNAlater for future projects.

- Line 154: you may want to check this IACUC reference. From the outside looking in, it seems odd to have a protocol under the name of someone not involved in this study. Of course, there could many explanations. Simply a thought to double check.

Thank you for pointing this out. We had copied the protocol numbers from a previous manuscript, and accidentally included an extra protocol number, which was not used for this study. We have fixed this error.

- Line 156 and throughout: you did not measure fungal growth. You measured the amount of fungal DNA collected with a swab of the same bats at two points in time. This isn't growth, and presenting it as such could be confusing. The bats can spread the fungus among themselves and hibernacula serve as reservoirs. Because bats are constantly moving about hibernacula in winter and coming into contact with each other and new areas of the hibernaculum, calling changes in loads growth is not well defended.

In response to comments from Reviewer 1, we have incorporated several citations from the chytrid system. There are many papers from that system that define change in fungal loads as fungal growth (e.g., Sauer et al. 2018 Proc. R. Soc. B.). To be consistent with that published work, and to maintain clarity for readers, we have maintained the term fungal growth, but we have added units to all figures to indicate that we are referring to changes in ng of DNA from November to March.

- Lines 184-185. The assumption that not re-encountering a bat in late winter means the animal has died needs to be supported. As described above, bats vary in their timing of arrival at and departure from the hibernacula, and this activity might include time points after the first survey and before the last (see citation below). I think your assumption needs to be put in the context of this knowledge. Furthermore, bats do move among hibernacula during winter. This is a very poorly studied phenomenon for obvious reasons, but early studies clearly show this occurs. And more recent studies show bats are seen in different hibernacula among years, without clear indication of when the movement occurred.

We agree that bats could potentially move among sites in the middle of winter, although we found these movements to be undetectably rare among the banded bats in this study, and generally rare in most studies (e.g. Norquay et al 2013 J of Mammalogy, Davis and Hitchcock 1965 J of Mammalogy). To address this, we ran a simulation where we randomly selected 52.5% of bats for simulated mortality (the proportion that we did not recapture across all sites and years), regardless of temperature. Running that simulation 10,000 times, we never once found a significant negative effect of early fungal loads and a significant positive interaction between loads and temperature, as we do in our observed data. On average, the probability that random death or emigration unrelated to loads and temperature could produce the observed relationships is <0.1% ($P < 0.000625$). Therefore, it would be nearly impossible to produce the relationship we observed if large numbers of bats were leaving sites solely due to factors unrelated to temperature (e.g., normal emigration, disturbance from banding). We have included a graph from this simulation here and would be happy to add this to the paper as well, if the editor and reviewers prefer.

For each run in the simulation, we checked to see if four criteria were met: a negative effect of fungal loads, a significant effect of fungal loads, a positive interaction term, and a significant interaction term. This graph shows how many simulations (out of 10,000) met 1, 2, 3, or 4 of the criteria.

Finally, bats move about hibernacula and the efficacy of surveys is not 100%. You actually show this in your supplemental methods where you mention finding bats in later years that were previously missed.

We moved the relevant text from the supplement to the main text. This text explains that only 5% of banded bats that were not recaptured were found later (and always in the same hibernaculum), and that including them as recaptured or not recaptured in a given year did not affect the results of our analysis, suggesting that this was not a significant source of noise in our data. Again, we want to note that even though efficacy was not 100% (it never is in mark-recapture studies, and we have much higher recapture rates than most studies), this source of noise would only make it more difficult to detect the significant relationship between recaptures and temperatures that we observe.

Thus, I think this assumption is far over-stated as being "likely". An exception to this would be during the mass mortality period associated with WNS arrival, but I would not apply that to the years earlier or later. This does not mean that your data can't be analyzed similar to how they currently are. However, I think that similar to fungal growth rates, these data need to be presented as representing what they actual are. Recapture rates.

We now use "recapture rates" instead of "mortality rates" or "survival rates" throughout when discussing our methods and results. We have left the justification for why recapture rates are a good proxy for mortality rates in this system, but as we describe above, we have also clarified why our sampling schedule justifies this and what other sources of noise could explain the unexplained variation in our recapture rates.

- Line 214. Eliminate this section. There is so much more about your analyses in other sections that this seems odd to me.

We deleted the section heading that read “Statistical analyses”, because this provided an odd header for a relatively small component of text, but we have retained the citations.

- Discussion. The only WNS paper you discuss here are from the Langwig lab. I do not object to self-citations, that would be unfair and hypocritical, but others have found similar results as you, and yet those studies are simply ignored. This limits the contribution of your work because it places it in a vacuum. For example, you cite Langwig 2016 to support the idea of management of hibernacula temperatures. Yet, that 2016 paper only discusses that idea, and does not present data on the matter. Those data and studies do exist (see Johnson et al 2016 and Richter et al 1993 below).

As described above, we have added references regarding general bat hibernation ecology and WNS throughout.

In close, I have tremendous respect and appreciation for the valuable data presented in this manuscript. But I also think the manuscript could be tremendously improved by putting the study into context and considering the work that others have done. How are you results similar? How are they unique? This is more useful for the larger community. The WNS landscape in North America is large and different regions will have unique stories and species. All that is missed when we fail to recognize the contributions of others. Equally important, I do think the findings need to be presented in a more balanced way (recaptures are not survival, fungal loads are not fungal growth). In this way the contribution will be more readily apparent and have greater impact on the field.

Thank you for your support.

References

- Boyles, J. G., Boyles, E., Dunlap, R. K., Johnson, S. A., & Brack, V. (2017). Long-term microclimate measurements add further evidence that there is no “optimal” temperature for bat hibernation. *Mammalian Biology*, 86(1), 9-16.
- Boyles, J. G., Johnson, J. S., Blomberg, A., & Lilley, T. M. (2020). Optimal hibernation theory. *Mammal Review*, 50(1), 91-100.
- Grieneisen, L. E., Brownlee-Bouboulis, S. A., Johnson, J. S., & Reeder, D. M. (2015). Sex and hibernaculum temperature predict survivorship in white-nose syndrome affected little brown myotis (*Myotis lucifugus*). *Royal Society open science*, 2(2), 140470.
- Humphrey, S. R., & Cope, J. B. (1976). Population ecology of the little brown bat, *Myotis Lucifugus*, in Indiana and north-central Kentucky. *American Society of Mammalogists*.

Johnson, J. S., Reeder, D. M., McMichael III, J. W., Meierhofer, M. B., Stern, D. W., Lumadue, S. S., ... & Kath, J. A. (2014). Host, pathogen, and environmental characteristics predict white-nose syndrome mortality in captive little brown myotis (*Myotis lucifugus*). *PLoS One*, 9(11).

Johnson, J. S., Scafani, M. R., Sewall, B. J., & Turner, G. G. (2016). Hibernating bat species in Pennsylvania use colder winter habitats following the arrival of white-nose syndrome. *Conservation and Ecology of Pennsylvania's Bats*. Bradford: The Pennsylvania Academy of Sciences, 181-199.

Lilley, Thomas Mikael, et al. "White-nose syndrome survivors do not exhibit frequent arousals associated with *Pseudogymnoascus destructans* infection." *Frontiers in zoology* 13.1 (2016): 12.

Lilley, T. M., Anttila, J., & Ruokolainen, L. (2018). Landscape structure and ecology influence the spread of a bat fungal disease. *Functional ecology*, 32(11), 2483-2496.

Norquay, K. J. O., & Willis, C. K. R. (2014). Hibernation phenology of *Myotis lucifugus*. *Journal of Zoology*, 294(2), 85-92.

Richter, A. R., Humphrey, S. R., Cope, J. B., & Brack Jr, V. (1993). Modified cave entrances: thermal effect on body mass and resulting decline of endangered Indiana bats (*Myotis sodalis*). *Conservation Biology*, 7(2), 407-415.

We now include all of these references, including those that were involved in the citation software error in our first submission and did not show up in the references section.

Reviewer #3 (Remarks to the Author):

This very interesting manuscript synthesizes an impressive set of field data relating to bats hibernating in sites while parasitized by the fungus that causes white-nose syndrome. The introduction section sets up the story in the context of ecological traps and I was quite frankly excited to see the evidence and conclusions by the end of the introduction (despite is shallow depth of literature review regarding bat hibernation and WNS beyond the author line).

We thank the reviewer for the comments. We have added additional supporting citations regarding general bat ecology and WNS into the introduction.

Try as I might, after reading through the methods, results, discussion sections, I was not convinced the promise implied by the title and intro was fulfilled. Assumptions in an ecological inference study of this nature are everything, and I wasn't convinced that the assumptions of this study were met.

First, no convincing evidence is presented or cited to clearly demonstrate that the quantity of fungal gene fragments present on the swabbed wings of bats at any given time is directly and

closely associated with disease severity in that individual. Fungal particles on the surfaces of gregarious animals in roosts where the pathogen exists in the environment too seems unlikely to be closely coupled as an index of individual health.

Thank you for drawing our attention to this. Previous work has demonstrated that fungal loads are correlated with disease outcomes (McGuire 2016 EcoHealth, Langwig 2016 Phil Trans), and we have now clarified this in the text. In this manuscript, we also find that fungal loads are correlated with recapture success, which is additional supporting evidence that fungal loads are an index of bat health with WNS.

Second, the assumption that bats not found after banding was a good proxy for overwinter survival seems like a stretch, because bats can abandon sites due to banding disturbance or simply move elsewhere.

We thank the reviewer for their comments, which have helped strengthen the manuscript. Below we address a different comment regarding banding disturbance. Here we describe a simulation that we conducted to address the concern that banding disturbance might somehow affect our results (also described above). We ran a simulation where we randomly selected 52.5% of bats for simulated mortality (the proportion that we did not recapture across all sites and years), regardless of temperature. Running that simulation 10,000 times, we never once found a significant negative effect of early fungal loads and a significant positive interaction between loads and temperature, as we do in our observed data. On average, the probability that random death or emigration unrelated to loads and temperature could produce the observed relationships is <0.1% (0.000625). Importantly, any factors that could cause us not to recapture a bat that actually survived would only make it more difficult to detect a significant relationship between recaptures and temperatures, and it would be nearly impossible to produce the relationship we observed if bats left sites solely due to factors unrelated to temperature (e.g., emigration, disturbance from banding).

Furthermore, banding was discontinued in North America for several decades due to known survival effects, so it isn't possible to rule out the possibility that it was the banding that altered the very overwinter survival being measured.

We agree that the old banding techniques were harmful to bats. However, we want to note that in our colder sites, we have nearly 100% recapture success, which would not occur if our banding method was resulting in mortality or forcing bats to leave sites. The effect of banding was also equal across all temperatures, so even if there were an effect, it should not bias our results, as evidenced by the simulation mentioned above. In addition, banding has been widely used in survival studies on bats (Frick 2010 Science, Frick 2010 Journal of Animal Ecology, Maslo 2015 Conservation Biology). Also, other work has found no effects of modern banding methods on bat body condition (Locatelli 2019 Acta Chiropterologica), suggesting that less invasive modern methods are having minimal impacts on bats.

Third, all of these measures seem to assume that the bats stayed put in the hibernation sites throughout winter and that the temperatures measured by the data loggers were those the bats

actually experienced. Individual bats can move during winter and where you find them during period checks may not reflect the temperature conditions they were seeking before or after that snapshot (single sampling period) of time.

We completely agree that bats can move within hibernacula during the winter and that temperatures fluctuate within a hibernaculum throughout hibernation. We have clarified both of these points in our manuscript. We have also moved some of these details from the supplement to the main text. However, our analysis is not invalidated by bats moving within hibernacula or temperatures changing within hibernacula between November and March. The point temperatures we observed for individual bats (measured by laser thermometer) were highly correlated with average November temperature in a given section within a given hibernaculum (measured by reference loggers)(Fig. S1), so we know that our point estimates were representative of that early period within the hibernation season. We think temperatures during that early period are critical for determining infected bat survival, because the sooner that bats start to experience higher fungal loads and disruptive hibernation physiology, the faster they will burn through their winter energy stores. We have now clarified this in the manuscript. We also show that fungal growth rates from November to March on recapture bats were not correlated with March temperatures; it is the early temperatures that predict bat disease outcomes.

Fourth, temperature is the only microclimate variable considered, even though prior work with WNS and hibernating bats built a large body of evidence (empirical and theoretical) indicating temperature interacts dynamically with humidity to influence survival. There also seems to be little consideration of seasonally dynamic changes in temperature within the hibernation sites and the options individual bats have to move among sites during winter to fit their thermoregulatory needs.

We thank the reviewer for this comment and agree that humidity likely plays an important role in WNS dynamics. Unfortunately, commercially available instruments used to measure humidity within hibernacula are too crude to capture fine-scale differences among individuals that are likely important for host-pathogen biology. Loggers deployed in hibernacula typically become saturated and fail to read accurate measurements above 90%. We recently designed a new psychrometer that can more accurately capture humidity variation relevant for both the pathogen and bats, and we are hoping to be able assess the effects of humidity in future analyses. Unfortunately, there is no existing individual-level data that can be used to understand the effects of humidity. However, the

exclusion of humidity from these analyses do not detract from the importance of temperature on fungal growth and bat recaptures.

Maybe I missed something, but the metrics used to build the case for bats funneling into ecological trap conditions during winter just don't seem suited for the rock-solid kind of evidence needed to support such a claim.

We hope that our updated manuscript clarifies these points for you and future readers.

REVIEWER COMMENTS

Reviewer #1 (Remarks to the Author):

The authors have thoroughly revised the manuscript and have addressed the concerns of all the reviewers in great detail. I have no further comments and recommend that this manuscript be accepted.

Reviewer #2 (Remarks to the Author):

Dear Hopkins and Colleagues,

I enjoyed reading your revised manuscript, "Ecological traps and thermal refugia mediate host survival with an infectious disease". Your manuscript contains a unique and valuable dataset collected over several years before and after the arrival of WNS in a large study area. The manuscript contains valuable data on habitat use by bats and fungal loads during this time, providing important insight into how bats are responding to this catastrophic disease. You have truly done an incredible amount of work in the field and in your analyses of the resulting data.

The manuscript argues for the presence of ecological traps and refuges from disease based on estimations of recapture rates, fungal loads, and habitat use. The data are compelling in several places and many of the insights into your responses to previous comments were quite helpful. However, most of these data are confirmatory of previous studies with the exception of the continued presence of bats in hibernacula with temperature >8 C, which I think should be clarified (see rationale below). Perhaps as an inescapable side-effect of peer-review, the manuscript has a great many supporting analyses and simulations, but I must confess I had difficulty getting to some of the key data needed to lend credence to the title and its promised result. Thus, I respectfully have misgivings about the manuscript. This is not to say that the data are not tremendously valuable. Only that I am unconvinced in the area that the manuscript claims to have a unique contribution.

A large point of confusion for me is that there is a mixed message in this manuscript regarding the data representing proof of the ecological trap (where bats were found). Please allow me to set aside the long-established fact that WNS effects are increased with temperature, which was established excellently by the PI of this manuscript in their 2012 publication in Ecology Letters. I say we should put this aside because if temperature-dependent decline during the WNS invasion (aka, mortality) phase is enough to declare an ecological trap, then this has been established for about 8 years now. Instead, I believe we should focus on whether or not bats are continuing (emphasis on continuing) to select warm environments despite the risk. For this manuscript to have a new contribution, I believe this question needs to focus on the post-invasion (aka, endemic) period that the authors define as >2 years after the arrival of the fungus. I believe this is the authors intent because they write in their response letter:

"In contrast [to a previous study showing movement to colder hibernacula AND colder rooms within hibernacula], our dataset tracks individual bats to determine their microclimate use, infection loads, and disease outcomes from pre- to post invasion, and shows that bats are continuing to prefer and use the warm sites where pathogen growth rates are highest and recapture rates are lowest".

Again, an emphasis on continuing.

However, the manuscript casts this timing differently in the Discussion and Results. In the Discussion (L.319):

"during and after fungal invasion, most bats (74%) continued to hibernate in relatively warm microsites ($>8^{\circ}\text{C}$) or relatively warm hibernacula (e.g., Fig. 1B)."

In the Results (L. 273-276):

"Across 12 hibernacula that were surveyed before (N=455 sampled and 6213 counted bats), during (N=439 sampled and 2078 counted bats), and after (N=176 sampled and 361 counted bats) fungal invasion, a large proportion (74%) of the few bats persisting in the region during post invasion years still used relatively warm roosts (i.e., $>8^{\circ}\text{C}$)."

So, to be clear, the Discussion states that basis for bats being trapped in the warm environment (74% being in those sites) is drawn on the invasion AND post-invasion stage. Inclusion of the invasion stage muddies the waters here, as those bats were behaving under the "old rules" of the costs and benefits of torpor and hibernation that WNS has changed. I do not think it merits the presence of an ecological trap. Further, as discussed in your response letter, it is after the invasion stage that is interesting. And previous work in Pennsylvania shows a shift in habitat selection, so putting these two time points together in the generation of a percent is potentially misleading. Now, the results contradict this and are more in line with what I just suggested is the important frame of reference: post-invasion.

So, I think an important question is this: were 74% of the 361 bats counted after fungal invasion (i.e., >2 years after Pd arrival) found in hibernacula warmer than 8°C ? Or does the 74% include the invasion period? If the former, that is very interesting indeed! It also begs the question, was the annual count continuing to decline throughout the post-invasion period (looks like years 3 and 4 on your figures?)? If 74% of bats are stuck in these traps, one would expect the decline to be continuing at a rapid pace, but I saw no mention of trends within the post-invasion period.

This needs to be more clearly defined for the reader. In my opinion, it is currently hard to follow all the ways in which you curate your data to select what will be used in analyses and why. I simply cannot see the forest for the trees here.

On a similar point, I do not follow the justification for not using bats that tested negative for Pd in your "growth rate" analysis. On L623-625 you say you used data only from bats that had Pd on them during early hibernation to avoid focus on pathogen transmission. Curating your data in this way does not take pathogen transmission out of the equation. As one of the authors of your manuscript recently showed, bats are capable of spreading the disease to bats hibernating elsewhere in the hibernacula. The environment itself has been shown to be a reservoir for the fungus. Thus, changes in fungal loads between two points in time is a product of more than just fungal growth. The authors therefore should call their measurement what it is: "change in fungal load". And it would have to include bats that tested negative during early hibernation. The justification above just doesn't make sense given what is known about how bats acquire Pd. All of the modeling of fungal growth rates and associated assumptions (e.g., 174-192) therefore seem out of place.

I appreciate most of the other changes, especially replacing survival with recapture rates. However, you do conflate some things when you say: "Bats experiencing WNS symptoms often leave their hibernacula (Frick et al. 2016), and bats that leave hibernacula mid-winter are likely to die from exposure or starvation on the landscape (Humphrey et al. 1976)." The emergence of dying bats during winter in the midst of the WNS mortality phase is well-known. But that is not the case of bats that are not suffering from severe WNS, as this sentence implies. Simple proof of that can be seen from searching for the literature bat activity in winter, which includes *Myotis* species, even in areas with more severe winters than your study area. While little browns are less well-known for this than other species, the small-footed bat (both eastern and western) are smaller and routinely are active aboveground on cold winter nights. Thus, bats switch hibernacula during winter. As cited in the last review, this is established and shouldn't be ignored as, "those bats die". These movements should be given acknowledged in your discussion. You mentioned in your response

letter that, "...it would be nearly impossible to produce the relationship we observed if large numbers of bats were leaving sites solely due to factors unrelated to temperature (e.g., normal emigration, disturbance from banding)." However, this does not negate the fact that winter movements and failure to re-sight should not be assumed as mortality. Stated differently, my point here is not that the results you documented are due to factors other than temperature, but that movement in response to temperature occurs in bats and does not necessarily mean the bats died. The Boyles et al. 2007 paper you cite demonstrates this (although within a hibernaculum). You mention that only 5% of bats were missed (L.212), but this is only based on what you know. It is unlikely that you later found all of the bats that you missed. Regardless, this suggests that either that bats can be missed during surveys or that bats move among hibernacula and do not die. Regardless, I think you have too simply described winter ecology here (also, see Boyles new paper on Optimal Hibernation Theory, as in the Discussion you discuss optimal temperatures).

There are other small examples where extraneous information is included and only adds confusion. For example, L. 159-160:

"Finally, we banded all swabbed bats that could be safely removed from their roosts (mean=73%, median = 100% of all swabbed bats) with an aluminum lipped-band.."

Here, I think the median and mean pertain to different datasets because if the mean is <100%, how can the middle value (median) be 100%? Overall, I recommend carefully checking all of the numbers presented throughout the manuscript. Given the sheer number of them, it is difficult to avoid small mistakes like these.

All the best in your effort and thank you for your hard work on this topic.

REVIEWER COMMENTS

Reviewer #1 (Remarks to the Author):

The authors have thoroughly revised the manuscript and have addressed the concerns of all the reviewers in great detail. I have no further comments and recommend that this manuscript be accepted.

Thank you!

Reviewer #2 (Remarks to the Author):

Dear Hopkins and Colleagues,

I enjoyed reading your revised manuscript, "Ecological traps and thermal refugia mediate host survival with an infectious disease". Your manuscript contains a unique and valuable dataset collected over several years before and after the arrival of WNS in a large study area. The manuscript contains valuable data on habitat use by bats and fungal loads during this time, providing important insight into how bats are responding to this catastrophic disease. You have truly done an incredible amount of work in the field and in your analyses of the resulting data.

Thank you! We appreciate your feedback.

The manuscript argues for the presence of ecological traps and refuges from disease based on estimations of recapture rates, fungal loads, and habitat use. The data are compelling in several places and many of the insights into your responses to previous comments were quite helpful. However, most of these data are confirmatory of previous studies with the exception of the continued presence of bats in hibernacula with temperature >8 C, which I think should be clarified (see rationale below). Perhaps as an inescapable side-effect of peer-review, the manuscript has a great many supporting analyses and simulations, but I must confess I had difficulty getting to some of the key data needed to lend credence to the title and its promised result. Thus, I respectfully have misgivings about the manuscript. This is not to say that the data are not tremendously valuable. Only that I am unconvinced in the area that the manuscript claims to have a unique contribution.

Below, we address the miscommunication that led to some of the confusion about bat behavior and disease outcomes during the post-invasion period. To help clarify which data contributed to each analysis, we created a flow diagram for the supplement that shows which bats went into each analysis and why. This diagram lists all of the sample sizes for each analysis, broken down by the pre-invasion, invasion, and post-invasion periods. We think this new figure will help interested readers to better understand the fine details of our analyses, so we thank the reviewer for pointing out the need to clarify things.

A large point of confusion for me is that there is a mixed message in this manuscript

regarding the data representing proof of the ecological trap (where bats were found). Please allow me to set aside the long-established fact that WNS effects are increased with temperature, which was established excellently by the PI of this manuscript in their 2012 publication in Ecology Letters. I say we should put this aside because if temperature-dependent decline during the WNS invasion (aka, mortality) phase is enough to declare an ecological trap, then this has been established for about 8 years now. Instead, I believe we should focus on whether or not bats are continuing (emphasis on continuing) to select warm environments despite the risk. For this manuscript to have a new contribution, I believe this question needs to focus on the post-invasion (aka, endemic) period that the authors define as >2 years after the arrival of the fungus. I believe this is the authors intent because they write in their response letter:

"In contrast [to a previous study showing movement to colder hibernacula AND colder rooms within hibernacula], our dataset tracks individual bats to determine their microclimate use, infection loads, and disease outcomes from pre- to post invasion, and shows that bats are continuing to prefer and use the warm sites where pathogen growth rates are highest and recapture rates are lowest".

Again, am emphasis on continuing.

However, the manuscript casts this timing differently in the Discussion and Results. In the Discussion (L.319):

"during and after fungal invasion, most bats (74%) continued to hibernate in relatively warm microsites (>8°C) or relatively warm hibernacula (e.g., Fig. 1B)."

In the Results (L. 273-276):

"Across 12 hibernacula that were surveyed before (N=455 sampled and 6213 counted bats), during (N=439 sampled and 2078 counted bats), and after (N=176 sampled and 361 counted bats) fungal invasion, a large proportion (74%) of the few bats persisting in the region during post invasion years still used relatively warm roosts (i.e., >8°C)."

So, to be clear, the Discussion states that basis for bats being trapped in the warm environment (74% being in those sites) is drawn on the invasion AND post-invasion stage. Inclusion of the invasion stage muddies the waters here, as those bats were behaving under the "old rules" of the costs and benefits of torpor and hibernation that WNS has changed. I do not think it merits the presence of an ecological trap. Further, as discussed in your response letter, it is after the invasion stage that is interesting. And previous work in Pennsylvania shows a shift in habitat selection, so putting these two time points together in the generation of a percent is potentially misleading. Now, the results contradict this and are more in line with what I just suggested is the important frame of reference: post-invasion.

So, I think an important question is this: were 74% of the 361 bats counted after fungal invasion (i.e., >2 years after Pd arrival) found in hibernacula warmer than 8 C? Or does

the 74% include the invasion period? If the former, that is very interesting indeed! It also begs the question, was the annual count continuing to decline throughout the post-invasion period (looks like years 3 and 4 on your figures?)? If 74% of bats are stuck in these traps, one would expect the decline to be continuing at a rapid pace, but I saw no mention of trends within the post-invasion period.

This needs to be more clearly defined for the reader. In my opinion, it is currently hard to follow all the ways in which you curate your data to select what will be used in analyses and why. I simply cannot see the forest for the trees here.

We thank the reviewer for this valuable feedback. While our analysis of early hibernation roosting distributions does compare post-invasion only to pre-invasion only distributions, in trying to summarize this pattern succinctly in the discussion, we reported a single summary statistic for the proportion of bats roosting above 8°C after year 0, which you correctly point out lumps the invasion and post-invasion years. We apologize for the confusion this caused! As Figure 5 shows, there's a nearly bimodal distribution during post-invasion, where many bats are still using warmer roosts, despite an average shift towards colder roosts. We have now edited the relevant sentences to report the proportion of bats roosting >8°C during post-invasion years only, which is still >50% of bats.

We have also added a new analysis that further addresses your concerns about whether bats are continuing to prefer warm roosts and whether populations are continuing to decline in warm roosts during the post-invasion period. In this new analysis, we calculated two bat population growth rates for each hibernaculum that was sampled during the post-invasion period. One population growth rate reflected the over summer change in population size from March to November, and is thus a proxy for immigration and recruitment into a site each year. The other population growth rate reflected the over winter change in population size from November to March, and is thus a good proxy for over winter mortality (and possibly emigration). As expected based on the temperature-dependent recapture data, we found a negative relationship between the average November roosting temperature in a hibernaculum and overwinter population growth rates, consistent with more bats dying in the warmest sites during the post-invasion years. Furthermore, there is a marginally significant interaction between the demographic season (over summer or over winter) and temperature, because over summer immigration rates tended to *increase* with average roosting temperature. Some noise is added into this analysis by aggregating across individual bat behavior at the hibernaculum level, but we think this new analysis complements our individual-level analysis and helps to illustrate bat preferences in another way. The new figure is now included in our supplement.

On a similar point, I do not follow the justification for not using bats that tested negative for Pd in your "growth rate" analysis. On L623-625 you say you used data only from bats that had Pd on them during early hibernation to avoid focus on pathogen transmission. Curating your data in this way does not take pathogen transmission out of

the equation. As one of the authors of your manuscript recently showed, bats are capable of spreading the disease to bats hibernating elsewhere in the hibernacula. The environment itself has been shown to be a reservoir for the fungus. Thus, changes in fungal loads between two points in time is a product of more than just fungal growth. The authors therefore should call their measurement what it is: "change in fungal load". And it would have to include bats that tested negative during early hibernation. The justification above just doesn't make sense given what is known about how bats acquire Pd. All of the modeling of fungal growth rates and associated assumptions (e.g., 174-192) therefore seem out of place.

We have changed the relevant variable name to “change in fungal loads” throughout. To clarify, our goal is to remove the confounding effects of *the timing of first infection*: if bats were uninfected in November and were Pd positive in March, we would not have been able to determine when they had become infected. We did not want to confound our estimates of fungal load change between two time points (the data we had for most banded bats) by coupling it with data from bats which may have gotten infected at any time point between November and March. This was a minority of individuals as infection prevalence was >90% by November. We have now clarified this in the referenced sentence in the supplementary methods and in the new flow diagram in the supplement.

I appreciate most of the other changes, especially replacing survival with recapture rates. However, you do conflate some things when you say: "Bats experiencing WNS symptoms often leave their hibernacula (Frick et al. 2016), and bats that leave hibernacula mid-winter are likely to die from exposure or starvation on the landscape (Humphrey et al. 1976)." The emergence of dying bats during winter in the midst of the WNS mortality phase is well-known. But that is not the case of bats that are not suffering from severe WNS, as this sentence implies. Simple proof of that can be seen from searching for the literature bat activity in winter, which includes *Myotis* species, even in areas with more severe winters than your study area. While little browns are less well-known for this than other species, the small-footed bat (both eastern and western) are smaller and routinely are active aboveground on cold winter nights. Thus, bats switch hibernacula during winter. As cited in the last review, this is established and shouldn't be ignored as, "those bats die". These movements should be given acknowledged in your discussion. You mentioned in your response letter that, "...it would be nearly impossible to produce the relationship we observed if large numbers of bats were leaving sites solely due to factors unrelated to temperature (e.g., normal emigration, disturbance from banding)." However, this does not negate the fact that winter movements and failure to re-sight should not be assumed as mortality. Stated differently, my point here is not that the results you documented are due to factors other than temperature, but that movement in response to temperature occurs in bats and does not necessarily mean the bats died. The Boyles et al. 2007 paper you cite demonstrates this (although within a hibernaculum). You mention that only 5% of bats were missed (L.212), but this is only based on what you know. It is unlikely that you later found all of the bats that you missed. Regardless, this suggest that either that bats can be missed during surveys or

that bats move among hibernacula and do not die. Regardless, I think you have too simply described winter ecology here (also, see Boyles new paper on Optimal Hibernation Theory, as in the Discussion you discuss optimal temperatures).

We appreciate this suggestion and agree the bat hibernation ecology is complex, with additional complexity being further revealed as we learn more about WNS. We agree that even in the relatively simple mine hibernacula where we extensively search for bats, we do occasionally miss a bat. We also agree that it is possible that a few bats may be leaving our sites mid-winter, not dying, and never returning to the same site, despite strong site fidelity in bats. We further clarified general bat ecology and WNS ecology throughout and added text that recapture rates are not a perfect measure for survival.

There are other small examples where extraneous information is included and only adds confusion. For example, L. 159-160:

"Finally, we banded all swabbed bats that could be safely removed from their roosts (mean=73%, median = 100% of all swabbed bats) with an aluminum lipped-band.."

Here, I think the median and mean pertain to different datasets because if the mean is <100%, how can the middle value (median) be 100%? Overall, I recommend carefully checking all of the numbers presented throughout the manuscript. Given the sheer number of them, it is difficult to avoid small mistakes like these.

We have carefully checked all of the numbers throughout the manuscript and also added the new flow diagram to clarify the sample sizes. In the specific instance mentioned here, the numbers were correct. Based on a prior comment from a reviewer, we wanted to clarify that we banded most bats that we swabbed/sampled in most surveys; in most surveys, we banded 100% of the bats, but in some surveys, some of the bats were not removed from their roosts to band for a variety of reasons (e.g., flooding, bat roosting height). Just to clarify the math with a quick example, if we had only done three surveys where we banded 10%, 100%, and 100% of the bats, the mean would be 70% and the median would be 100%.

All the best in your effort and thank you for your hard work on this topic.
Thank you!